# Is A Picture Worth A Thousand Words? Delving Into Spatial Reasoning for Vision Language Models

**Jiayu Wang**[1]   **Yifei Ming**[2*]   **Zhenmei Shi**[1]   **Vibhav Vineet**[3]
**Xin Wang**[3]   **Yixuan Li**[1]   **Neel Joshi**[3]

[1]University of Wisconsin–Madison   [2]Salesforce AI Research
[3]Microsoft Research

{milawang,zhmeishi,sharonli}@cs.wisc.edu
yifei.ming@salesforce.com
{vibhav.vineet,wanxin,neel}@microsoft.com

## Abstract

Large language models (LLMs) and vision-language models (VLMs) have demonstrated remarkable performance across a wide range of tasks and domains. Despite this promise, spatial understanding and reasoning—a fundamental component of human cognition—remains under-explored. We propose SpatialEval, a novel benchmark that covers diverse aspects of spatial reasoning such as relationship understanding, navigation, and counting. We conduct a comprehensive evaluation of competitive language and vision-language models. Our findings reveal several counter-intuitive insights that have been overlooked in the literature: (1) Spatial reasoning poses significant challenges where competitive models can fall behind random guessing; (2) Despite additional visual input, VLMs often under-perform compared to their LLM counterparts; (3) When both textual and visual information is available, multi-modal language models become less reliant on visual information if sufficient textual clues are provided. Additionally, we demonstrate that leveraging redundancy between vision and text can significantly enhance model performance. We hope our study will inform the development of multimodal models to improve spatial intelligence and further close the gap with human intelligence. Our code is available at https://github.com/jiayuww/SpatialEval.

## 1 Introduction

The recent breakthroughs in foundation models have had a transformative effect on research and industry, and we have seen these models rapidly integrated into products and new businesses that are shaping people's lives for the better. This sea change was initially driven by large language models (LLMs), which have shown at times near unbelievable, human-level performance across a wide range of tasks. Over the past year, many of these models have been extended to handle images in addition to text, leading to a significant increase in vision-language models (VLMs), especially multimodal large language models (MLLMs), which demonstrate groundbreaking performance in image-related tasks as in the text domain.

However, the reality is not quite as rosy as advertised. While these models have been instrumental in advancing the state-of-the-art in complex reasoning tasks such as common sense reasoning, mathematical problem solving, and scientific question answering [17, 30, 41, 72, 75], they have not been as effective for a number of problem domains. In particular, as we will show in this paper, they have limited performance on tasks that require detailed visual understanding and reasoning of images.

---

*The work was completed in part during Yifei Ming's internship at Microsoft Research, as well as PhD thesis research at UW-Madison.

38th Conference on Neural Information Processing Systems (NeurIPS 2024).

Visual understanding and reasoning—an intrinsic part of human perception and cognitive ability—have been largely under-explored when it comes to LLMs and VLMs. In fact, it is often argued that the visual sense is the dominant sense in people, yet when it comes to current models, it seems quite secondary. Spatial reasoning, in particular, is fundamental to everyday human activities such as navigating environments, understanding maps, and manipulating objects. It encompasses skills that are crucial for both survival and higher-order cognition, including the ability to navigate through space, recognize patterns, and deduce relationships from spatial configurations.

In this paper, we propose **SpatialEval**, a novel benchmark containing four tasks (`Spatial-Map`, `Maze-Nav`, `Spatial-Grid`, and `Spatial-Real`, Section 2.1) to explore the performance of LLMs and VLMs on diverse aspects of spatial reasoning, including relationship, navigation, position understanding, and object counting. Humans excel at such tasks, making them essential capabilities for intelligent systems to emulate for safe and effective deployment in the real world.

Our dataset, however, is constructed with a key twist – each problem in our benchmark has an image and a text representation that is sufficient for answering each spatial understanding question. We denote the use of these sources as VQA, which is the standard task of visual-question answering that consists of a vision-only input and a question, TQA, text-only input and a question, and VTQA, a combination of the previous with vision and text input.

We conduct a systematic and comprehensive evaluation of a wide range of open-source and proprietary LLMs and VLMs. We perform in-depth analysis and unveil several novel and surprising results that challenge the current understanding of how these models process spatial information:

- **Spatial reasoning remains challenging**: VLMs frequently struggle with spatial reasoning tasks, with some competitive models performing worse than random guessing.
- **Visual vs. textual inputs**: Without detailed textual descriptions, multimodal models rarely surpass the performance of their LLM backbones when relying solely on visual inputs. This underscores the critical role of text in enhancing model performance on spatial reasoning.
- **Reduced reliance on visual information**: When both textual and visual inputs are provided, multimodal language models tend to rely less on the visual component if sufficient textual clues are available.
- **Textual performance of VLMs**: VLMs often outperform their LLM counterparts with text-only inputs, indicating that the language model backbones within VLMs benefit from multimodal training, despite a lack of similar benefits from the visual components.

We surmise the limitations in VLMs' spatial understanding stem from the overly simplistic handling of visual information in current architectures and training pipelines. We believe the contributions of this paper will drive changes in model design, accelerating improvements that could unlock more robust spatial reasoning capabilities, and help bridge the gap toward human-like intelligence.

## 2 Dataset and Task Construction

### 2.1 Dataset Setup

To evaluate the spatial reasoning abilities of LLMs and VLMs, we construct four diverse tasks including spatial relationships, navigation, position understanding, and counting. To systematically study the impact of modality, we design three types of input formats for each task: (1) TQA (Text-only): the input is purely textual and contains all necessary information for a person to answer the questions. (2) VQA (Vision-only): the input consists solely of an image, which provides sufficient details for a person to easily answer, a format also referred to as Visual Question Answering (VQA) in the literature. (3) VTQA (Vision-text): the input includes both an image and its textual representation with detailed descriptions, rendering the information in both modalities redundant. We evaluate LLMs using Text-only inputs and VLMs using Text-only, Vision-only, and Vision-text inputs on the same set of questions (Table 1). The synthetic tasks are curated based on the following key guidelines: (1) Avoidance of data leakage—since LLMs are pre-trained on web-scale data, it is crucial to ensure that the test data have not been seen during training; (2) configurability—being configurable allows for controlled experiments and extends easily to additional tasks; (3) scalability—the ability to scale the number of test samples enhances the statistical significance of experimental results.

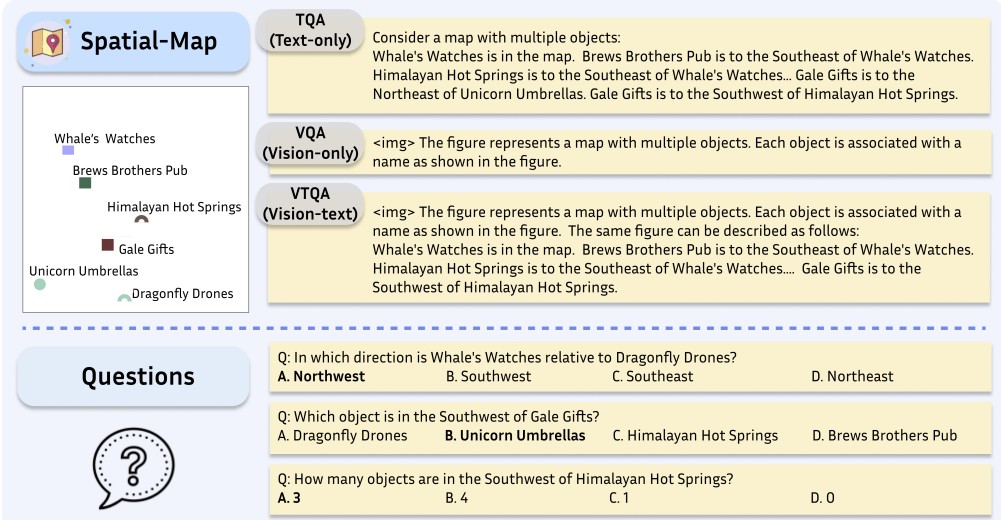

Figure 1: Illustration of the `Spatial-Map` task, which simulates a map with multiple locations. To investigate the impact of modality, we consider three input formats: Text-only, Vision-only, and Vision-text. We evaluate language models (w. TQA input) and vision-language models (w. VQA and VTQA inputs) on the same set of questions.

**Spatial-Map.** Understanding the spatial relationships among objects on a map is a fundamental aspect of human cognitive abilities. To simulate this environment, we create a map-like dataset termed `Spatial-Map` with $K$ objects, where $K$ is configurable. Each object is associated with a unique location name, such as Unicorn Umbrellas and Gale Gifts. To study the impact of modality, the textual representation of each input consists of pairwise relations such as `Brews Brothers Pub is to the Southeast of Whale's Watches`. An example with $K = 6$ is shown in Figure 1, with Text-only, Vision-only, and Vision-text inputs. These questions include asking about the spatial relationships between two locations and the number of objects that meet specific spatial criteria.

**Maze-Nav.** Navigation through complex spaces is essential for intelligent systems. To evaluate such abilities, we have developed a maze-like dataset named `Maze-Nav`. Visually, each sample can be

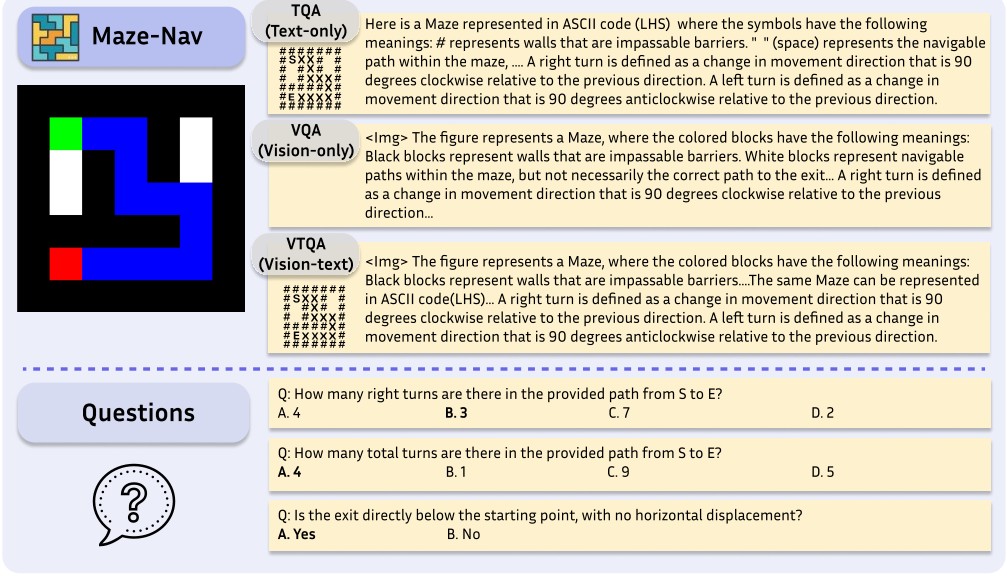

Figure 2: Illustration of the `Maze-Nav` task, which evaluates the model's ability to navigate from the starting point (S) to the exit (E).

represented as colored blocks where different colors signify distinct elements: a green block marks the starting point (S), a red block indicates the exit (E), black blocks represent impassable walls, white blocks denote navigable paths, and blue blocks trace the path from S to E. The objective is to navigate from S to E following the blue path, with movement permitted in the four cardinal directions (up, down, left, right). Alternatively, each input can be depicted in a textual format using ASCII code. An example is illustrated in Figure 2, featuring Text-only, Vision-only, and Vision-text inputs. We construct this task based on an open-sourced library [26]. The questions asked include counting the number of turns from S to E and determining the spatial relationship between S and E. While such questions are easy for humans, we will show in Section 3 that they still pose significant challenges for modern multimodal language models.

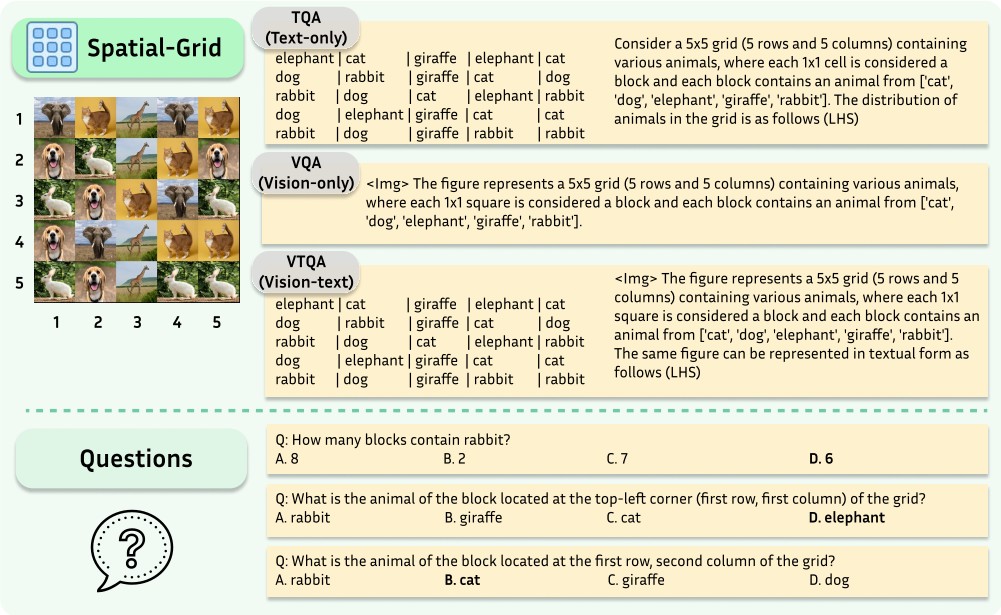

Figure 3: Illustration of the `Spatial-Grid` task, which evaluates the model's spatial reasoning ability in a rigid grid structure.

**Spatial-Grid.** To investigate spatial understanding within structured environments, we introduce a grid-like dataset named `Spatial-Grid`, contrasting with the `Spatial-Map` where objects are positioned arbitrarily. Visually, each input consists of a grid of cells, each containing an image (*e.g.,* a rabbit). An example is illustrated in Figure 3. Alternatively, this grid can also be represented in a purely textual format; for instance, the first row might be described as: elephant | cat | giraffe | elephant | cat. The evaluations focus on tasks such as counting specific objects (*e.g.,* rabbits) and identifying the object located at a specific coordinate in the grid (*e.g.,* first row, second column).

**Spatial-Real.** To extend the evaluation of spatial reasoning beyond synthetic environments, we introduce `Spatial-Real`, a task built on the Densely Captioned Images (DCI) dataset [67], where each image has a detailed caption with more than 1,000 words on average. As DCI does not contain questions, we curate multiple-choice questions regarding spatial reasoning (object counting, relation, and position understanding) and annotate the answers. An example is shown in Figure 4. We provide detailed analysis on `Spatial-Real` in Appendix E.

## 2.2 Models

We consider a wide range of competitive open-source language models with different scales, including Phi2-2.7B [35], the LLaMA family (LLaMA-2-7B, LLaMA-2-13B, and LLaMA-3-8B) [65], Mistral-7B [27], the Vicuna family (Vicuna-7B-1.5 and Vicuna-13B-1.5) [11], and Nous-Hermes-2-Yi-34B. For multimodal language models, we consider the Bunny family (Bunny-Phi-2-SigLIP, Bunny-Phi-1.5-SigLIP, Bunny-Phi-2-EVA, and Bunny-Phi-1.5-EVA) [22], CogVLM [69], CogAgent [23], InstructBLIP family (InstructBLIP-Vicuna-7B and InstructBLIP-Vicuna-13B) [13], and LLaVA family (LLaVA-1.6-Mistral-7B, LLaVA-1.6-Vicuna-7B, LLaVA-1.6-Vicuna-13B, and LLaVA-1.6-

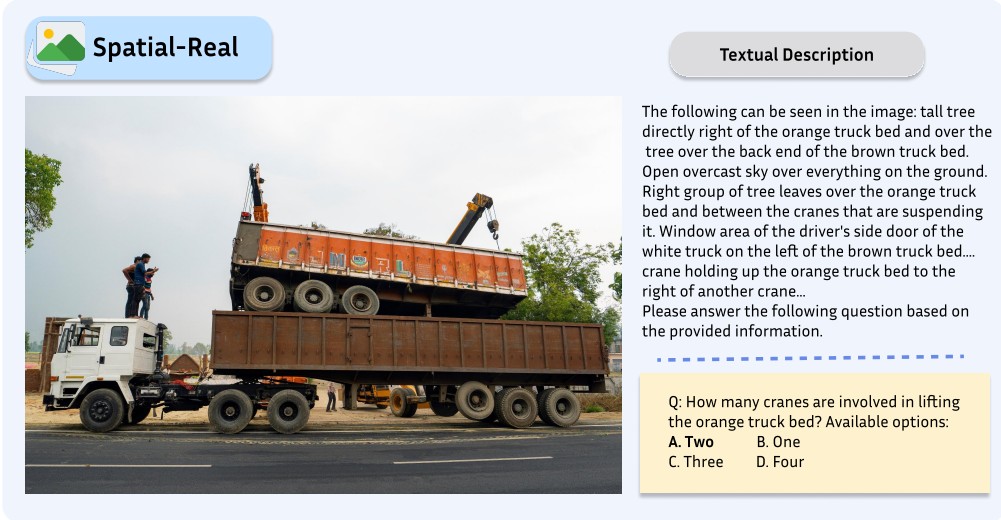

Figure 4: Illustration of the `Spatial-Real` task, which is built on real images with long captions, featuring detailed descriptions averaging over 1,000 words per image.

34B) [38]. We also evaluate the proprietary models: Open AI's GPT-4V, GPT-4o, GPT-4, Google Gemini Pro 1.0, and Anthropic Claude 3 Opus.

**Evaluation.** As each question contains four options, we use accuracy as the main evaluation metric. The same user prompt is appended at the end of each question: *First, provide a concise answer in one sentence. Then, elaborate on the reasoning behind your answer in a detailed, step-by-step explanation.* For each model, we adopt the default configurations and decoding strategies, *e.g.,* `argmax` for deterministic decoding and top-$p$ for non-deterministic decoding. For non-deterministic decoding, the results are averaged over three independent runs for open-source models. For proprietary models, due to their limited availability and increased compute time and cost, we only perform one run. We summarize the terminologies regarding input modalities for LLMs and VLMs in Table 1:

| Model | Input Modality | Term | Description |
|-------|----------------|------|-------------|
| LLM | Text-only | TQA (LLM) | Text-only input that includes all necessary information to answer questions without visual context. |
| VLM | Text-only | TQA (VLM) | Text-only input as in TQA (LLM) but applied to VLMs (*e.g.,* the LLaVA family). |
| VLM | Vision-only | VQA | Input only includes an image without corresponding textual description. |
| VLM | Vision-text | VTQA | Input includes both an image and its textual description. |

Table 1: Terms regarding input modalities for LLMs and VLMs.

We describe the Text-only, Vision-only, and Vision-text input modalities based on how we feed the image information to the models. Vision-only input means the image is fed directly to the models without textual description, while all questions are presented in text.

## 3 Main Results and Analysis

**Spatial reasoning remains surprisingly challenging.** The evaluation results on open-source models on `Spatial-Map`, `Maze-Nav`, and `Spatial-Grid` are shown in Figure 5. For each task, the reported accuracy is averaged over all questions. For vision-language models, we choose the Vision-only input format, commonly used in Visual Question Answering (VQA). We use a dashed red line in each figure to indicate the expected accuracy if answering by random guessing. Our findings reveal several notable insights: **(1)** Vision-only inputs: Despite the simplicity of these tasks for humans, most competitive multimodal models perform at levels similar to or barely above random guessing.

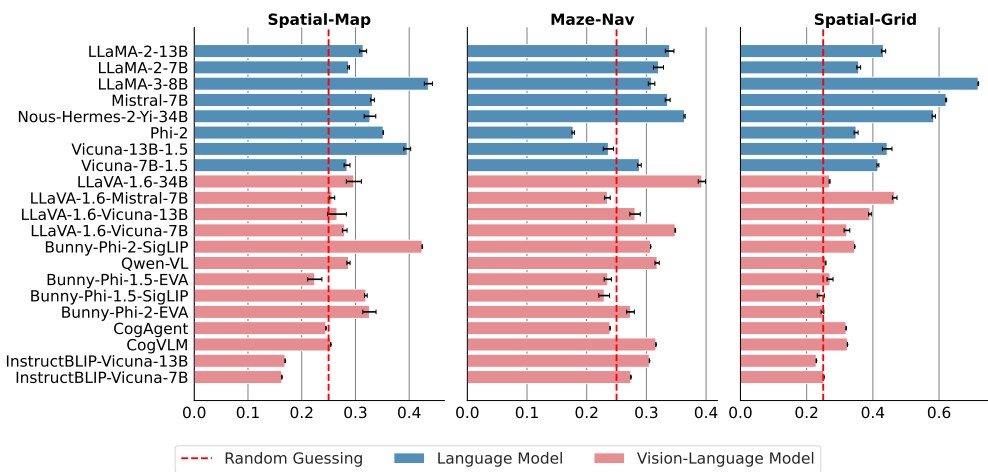

Figure 5: Performance overview on spatial reasoning tasks. We report the accuracy averaged over all questions. We consider the VQA (Vision-only) format for vision-language models. The dashed red line denotes the expected accuracy for random guessing. For `Spatial-Map` and `Maze-Nav` tasks, only a few models outperform random guessing by a notable margin.

**(2)** Text-only inputs: While the textual input includes essential spatial information, it generally does not significantly enhance the spatial reasoning capabilities of competitive models. An exception occurs in the `Spatial-Grid` task, where Llama-3 achieves an accuracy of 71.9%, followed by Mistral-7B-Instruct at 62.1%, both notably surpassing random guessing. Despite these successes, the performance of these models still lags significantly behind human levels. These results underscore the need for further development of techniques tailored to spatial understanding and reasoning.

**The impact of input modality.** To investigate the impact of modality, we compare the performance of a large language model (LLM) and a vision-language model (VLM) with the same language backbone. We consider the VQA (Vision-only) format for vision-language models. The results are shown in Figure 6. Each vertex on the spider plot represents the average accuracy of a (VLM, LLM) pair. We observe that on `Spatial-Map` and `Spatial-Grid`, the majority of VLMs yield worse performance compared to their LLM counterpart, despite having an additional visual encoder. For example, on `Spatial-Grid`, Mixtral-7B achieves an average accuracy of 62.1%, while LLaVA-v1.6-Mistral-7B only yields an accuracy of 47.1% (15% ↓). Detailed results can be seen in Appendix F.

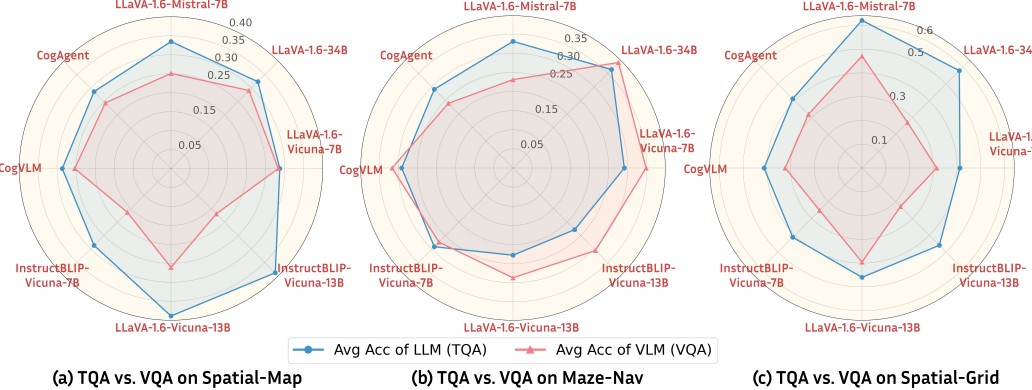

Figure 6: TQA (LLM) vs. VQA on spatial reasoning tasks. Each vertex on the spider plot represents the Avg Acc of a (VLM, LLM) pair with the same language backbone, i.e., LLM v.s. VLM further finetuned on that. VLMs are depicted in red, and LLMs in blue. We can see that VLMs rarely enhance the performance compared to their LLM counterparts.

# 4 Delving Into Spatial Reasoning for Vision-Language Models

## 4.1 Seeing Without Understanding: The Blindness of Multimodal Language Models

To better understand how VLMs process visual information, we conduct a series of controlled experiments in the VTQA (Vision-text input) setting. For each sample, we replace the original image input (that matches the textual description) with either: (1) *No Image*: only keep the textual input without the image input, (2) *Noise Image*: a Gaussian noise image irrelevant to the task, and (3) *Random Image*: a random image

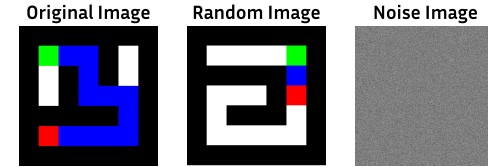

Figure 7: Illustration of *Random Image* and *Noise Image* for the example in Figure 2.

from the dataset that does not match the textual description, as shown in Figure 7.

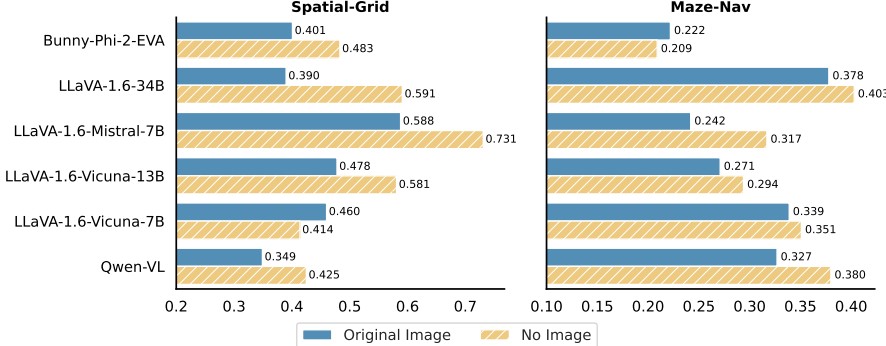

Figure 8: VTQA vs. TQA (VLM) on spatial reasoning tasks. VLMs exhibit improved performance in spatial reasoning tasks when visual input is absent.

**VLMs exhibit improved performance when visual input is absent.** We conducted experiments by entirely removing the *Original Image* and relying solely on the textual description. The results are shown in Figure 8. For each task, we report the accuracy averaged across all questions. Remarkably, the absence of visual input leads to better performance across a range of VLM architectures. For instance, the performance of LLaVA-1.6-34B on the `Spatial-Grid` task improved by 20.1% when no image was presented compared to scenarios with the original image. This observation underscores that when textual information alone can address the questions, additional visual inputs do not necessarily enhance, and may even hinder the performance, a sharp contrast to human capabilities where visual cues significantly aid in understanding. The removal of visual input forces the models to utilize the textual information to solve spatial reasoning tasks.

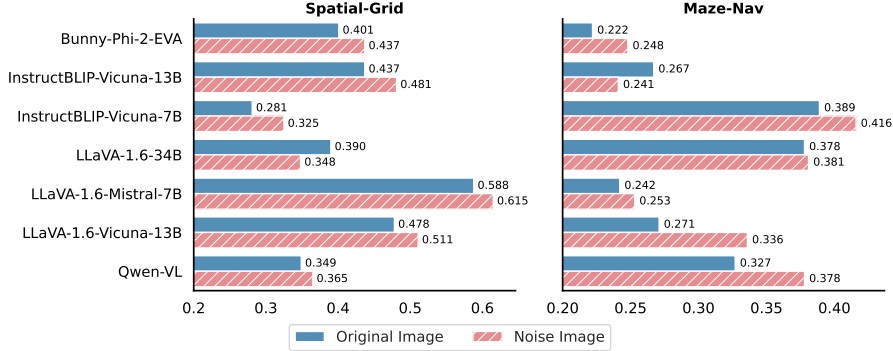

Figure 9: Original Image vs. Noise Image in VTQA. Replacing the original image with a Gaussian noise image improves the performance across diverse VLM architectures.

**Noise image can improve the performance.** We replace the *Original Image* with a *Noise Image* while retaining the original textual description. The results are shown in Figure 9. Consistent with

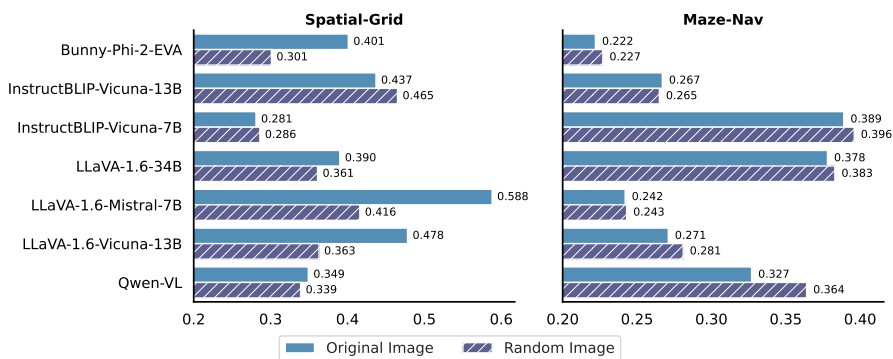

Figure 10: Original Image vs. Random Image in VTQA. On `Maze-Nav`, replacing the original image with a random image leads to performance improvement across diverse VLM architectures.

the findings in Original Image vs. No Image, using a noise image also improves the performance across various VLM architectures. For example, the accuracy of LLaVA-1.6-Vicuna-13B increases by 6.5% on the `Maze-Nav` task when the noise image is used as opposed to the original image. In contrast to the *No Image* setting, noise images provide limited visual cues. Nonetheless, the model tends to prioritize the textual information, especially when visual cues are not pertinent to the task.

**Mismatched image-text does not necessarily hurt.** To build on previous findings, we further investigate the effects of replacing the *Original Image* with a *Random Image* (illustrated in Figure 7). Unlike a noise image, a random image is task-related but may provide conflicting information compared to the textual description. Intuitively, one might expect that such random images would degrade VLM performance due to contradictory cues. However, as demonstrated in Figure 10, this expectation does not always hold true. For instance, *Random Image* in the `Maze-Nav` task leads to improved performance across various VLM architectures. This outcome implies that VLMs are not heavily reliant on visual information, particularly when adequate textual clues are provided.

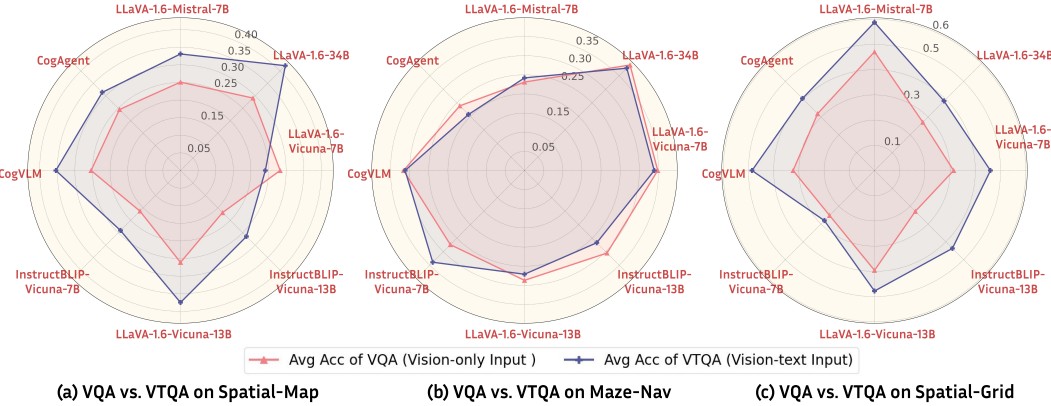

Figure 11: VQA vs. VTQA on spatial reasoning tasks. Each vertex on the spider plot represents the Avg Acc of a (Vision-only, Vision-text) pair with the same VLM model. We can see that having the additional textual input (VTQA) enhances the performance compared to only using images (VQA).

## 4.2 Leveraging Redundancy in Multimodal Inputs

Multimodal language models offer considerable versatility in handling multimodal inputs. While the visual input alone often provides sufficient details for humans to address spatial reasoning tasks with ease, we propose that VLMs significantly benefit from the inclusion of textual descriptions alongside visual data, even if this introduces substantial redundancy. We verify this hypothesis by comparing VQA (Vision-only input) and VTQA (Vision-text input) across diverse VLM architectures. The results are shown in Figure 11, where each vertex on the spider plot represents the average accuracy of a (Vision-only, Vision-text) pair based on the same VLM. For `Spatial-Map` and `Spatial-Grid`, we can clearly see that having the additional textual input (VTQA) enhances the performance compared

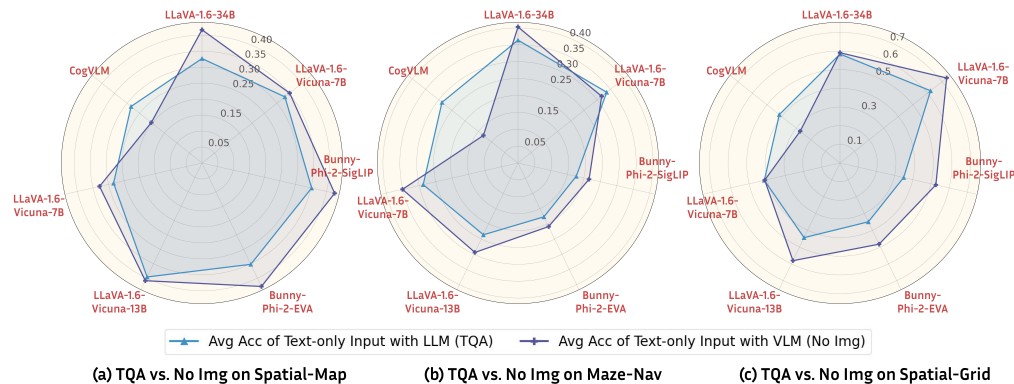

| (a) TQA vs. No Img on Spatial-Map | (b) TQA vs. No Img on Maze-Nav | (c) TQA vs. No Img on Spatial-Grid |

Figure 12: Comparison of Text-only input with LLM (TQA) vs. Text-only input with VLM (No Img). We consider VLMs that support text-only inputs. Each vertex on the spider plot represents the Avg Acc of a (LLM, VLM) pair with the same language model backbone.

to only using images (VQA) across different VLM architectures. This suggests that textual inputs improve the accuracy of spatial reasoning in VLMs. We further compare TQA and VTQA in Appendix D. Detailed results are included in Appendix F.

**Text-only input with LLM vs. Text-only input with VLM.** Given the demonstrated efficacy of text-only inputs, we conducted an ablation study to compare LLMs and VLMs using text-only inputs. We consider VLMs that are capable of processing text without accompanying visual data. The results are illustrated in Figure 12. Except for CogVLM, the majority of VLMs outperform their corresponding LLM backbones. This suggests that the language model backbones in VLMs demonstrate enhanced spatial reasoning abilities through multimodal learning. Conversely, the addition of visual information does not necessarily provide further benefits.

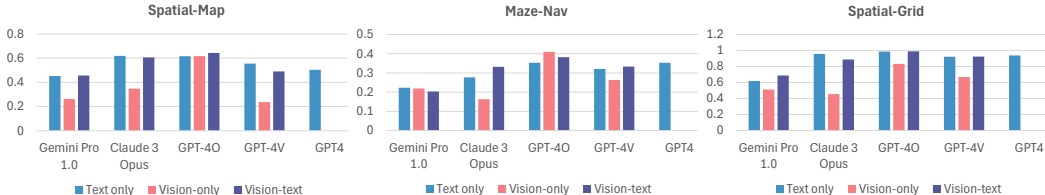

Figure 13: Results with proprietary models. Similar trends are observed as with open-source models.

## 4.3 Proprietary vs. Open-Source Models

As many recent benchmarks have shown that proprietary models generally outperform open-source models, it is important to understand if our observed trends hold with proprietary models. The performance of several top proprietary models (GPT-4, GPT-4V, GPT-4o, Gemini Pro 1.0, and Claude 3 Opus) are shown in Figure 13. We have the following salient observations: (1) A significant performance gap exists between SoTA open-source models and proprietary models, as expected. Furthermore, with both Text-only and Vision-text formats, GPT-4V and GPT-4o significantly outperform random guessing across all tasks. For instance, in the Vision-text format, GPT-4o achieves

| Comparison | Results | Summary of Findings |
|---|---|---|
| TQA (LLM) vs. VQA | Figure 6 | VQA rarely enhances the performance compared to TQA (LLM). |
| VTQA vs. TQA (VLM) | Figure 8 | VLMs exhibit improved performance in spatial reasoning tasks when the image input is absent. |
| VQA vs. VTQA | Figure 11 | Given the same image input, additional textual description enhances VLM's performance. |
| TQA (VLM) vs. TQA (LLM) | Figure 12 | Multimodal fine-tuning enhances LLM's spatial reasoning ability. |
| TQA (LLM) vs. VTQA | Figure 16 | No definitive winner. |

Table 2: Summary of main findings.

an accuracy of 0.989 on `Spatial-Grid` (Table 7). (2) Yet, the trends we observed with open source models hold, VQA consistently under-performs compared to TQA and VTQA, for example, GPT-4V's performance improves by 25.6% on `Spatial-Grid` when switching from Vision-only to Vision-text input; and again no clear winner between TQA and VTQA (see Appendix D for further details), showing that proprietary models, even the new GPT-4o model, still do not appear to fully leverage visual inputs. We summarize the key findings of this work in Table 2.

## 5 Related Work

**Large language models.** Large language models (LLMs) have achieved outstanding performance across a diverse array of fields, including finance [34], bioinformatics [63], law [60], education [29], coding [24], and creative tasks [2, 36]. LLM architectures have gone through significant changes in recent years, with notable developments such as BERT [14], OPT [76], PaLM [12], Gemma family [62], Mistral family [27], GPT family [2, 10], Claude family [5], and LLaMA family [3, 64, 66]. These models have demonstrated emergent abilities and revolutionized numerous domains, supporting capabilities such as in-context learning [46, 51, 59], compositional reasoning [16, 20, 70], and task-specific adaptation [57, 58, 71]. However, visual understanding and reasoning—an intrinsic part of human cognitive ability—remains largely under-explored for LLMs.

**Vision-language models and multi-modal language models.** The success of LLMs has propelled the adoption of the Transformer architecture [68] within the computer vision community, such as ViT [15], Beit [9], CLIP [54], MAE [21], Swin [42, 43], and DiT [53]. Building on the capabilities of powerful LLMs, multi-modal language models (MLLMs) such as Flamingo [4], LLaMA-Adapter [19, 74], LLava [37, 39], stable-diffusion [55], BLIP [32, 33], MiniGPT-4 [77], Qwen [7, 8], Gemini [61], MM1 [45] have significantly expanded the range of problems that can be addressed with improved reliability [49]. These models adeptly handle inputs from diverse modalities and have demonstrated remarkable performance on diverse tasks such as mathematical reasoning [44], image-text retrieval [47, 73], and visual reasoning [6, 18, 25, 28, 31, 40, 50].

**Spatial understanding and reasoning.** Spatial reasoning entails comprehending and manipulating spatial relationships, a task significantly more challenging than visual grounding [1, 28, 52]. Although progress in natural language processing, evaluations, and benchmarks on LLMs such as GPT-4 [2] and Claude3 [5] have predominantly focused on textual or relational reasoning. This focus often overlooks the intricate nature of spatial reasoning tasks. Notably, recent studies [17, 30, 41, 48, 56, 72, 75] have conducted thorough evaluations across a diverse range of tasks. Yet, they demonstrate a scant exploration of spatial reasoning capabilities, highlighting a gap in assessing this complex cognitive skill in current benchmarks. In particular, Zhang *et al.* [75] suggest that MLLMs primarily leverage textual cues rather than visual diagrams to solve math problems while focusing only on math problems. Yamada *et al.* [72] design simple navigation tasks and finds LLMs appear to capture certain aspects of spatial structure implicitly, but room for improvement remains, while our benchmarks are more complicated than their navigation tasks based on simple geometry maps. Fu *et al.* [17] propose a benchmark on science and games with limited exploration related to spatial reasoning, while we are dedicated diverse spatial reasoning tasks with in-depth analysis.

## 6 Discussion and Conclusions

We explored the spatial understanding capabilities of VLMs and LLMs. Our experiments resulted in several surprising conclusions across SoTA open-source and proprietary models. (1) VLMs struggle with spatial reasoning tasks, (2) multimodal models rarely surpass LLMs when relying on visual inputs, (3) when both textual and visual inputs are provided, multimodal language models rely less on the visual inputs, and (4) VLMs often outperform their LLM counterparts with text-only inputs. This challenges the belief that current VLMs are highly performant at a wide range of vision-text tasks. However, with further thought, perhaps this is to be expected after all. The currently known architectures for VLMs attempt to "translate" the vision input into the language space and all reasoning is then performed in the language domain. It is logical that this automatic translation path is worse than a human provided translation to text, as in our text-only scenario. Thus our work shows the limits of the translation approach. Instead, to bridge the gap to human performance, future models require new architectures that treat vision input as a first-class source of information and reason in a joint vision-language space. It is our hope that our work informs development on this path.

## Acknowledgement

The authors would like to thank NeurIPS anonymous reviewers for their insightful feedback and helpful discussions. Yifei Ming and Yixuan Li are funded in part by the AFOSR Young Investigator Program under award number FA9550-23-1-0184, National Science Foundation (NSF) Award No. IIS-2237037 & IIS-2331669, and Office of Naval Research under grant number N00014-23-1-2643.

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

# Appendix

## A  How Does SpatialEval Expand Beyond Traditional VQA?

We highlight several key rationales for SpatialEval in comparison to existing VQA benchmarks:

- **Task scope and focus**: Existing benchmarks such as Visual Genome [31], GQA [25], and BLINK [18] focus only on VQA, where the image is required but the text description is often omitted or optional. In contrast, SpatialEval further explores spatial reasoning across different settings: TQA (LLM), TQA (VLM), and VTQA, where images or texts can be optional, thereby broadening the scope of tasks.

- **Evaluation scheme**: we primarily focus on generative LLMs and VLMs which can "elaborate on the reasoning behind your answer in a detailed, step-by-step explanation" (Section 2.2). The task in prior VQA benchmarks is often treated as discriminative with no explicit reasoning. Therefore, it remains unknown if previous observations can be naturally transferred to foundation models pre-trained on web-scale data.

- **The Textual Representation of Images**: The textual descriptions in prior VQA benchmarks are often brief or directly imply the answers. While these datasets are valuable, they do not consistently offer the level of complexity we require, such as numerous objects containing dense visual information along with detailed natural language captions that fully convey the image content. In SpatialEval, we provide long and dense captions for each image. As a result, answers cannot be easily inferred. We also aim to isolate object detection capability from spatial reasoning ability by simplifying objects to symbols (*e.g.,* Spatial-Map).

- **IQ test for VLMs and LLMs**: Tasks in SpatialEval are designed to serve as cognitive tests that evaluate basic capabilities of multi-modal foundation models. While three tasks feature synthetic visual content, humans can solve them with near-perfect accuracy. This indicates that the tasks are within the realm of human cognitive capabilities.

## B  Limitations and Societal Impact

**Limitations.**  In this work, we introduce four novel tasks and conduct a comprehensive evaluation of diverse large language models (LLMs) and vision-language models (VLMs), presenting a controlled and thorough evaluation of spatial reasoning capabilities. While our analysis is detailed and extensive, it remains primarily empirical. We believe that embarking on a formal theoretical study, despite its challenges, would significantly enrich our understanding of pre-trained multimodal language models. Moreover, our focus has been on in-depth analysis rather than on developing new training strategies or adaptation algorithms to enhance spatial reasoning. Recognizing these points, we identify them as valuable directions for future research.

**Societal impact.**  Regarding the positive societal impact, our evaluations and observations could catalyze the development of new algorithms that enhance the spatial reasoning abilities of LLMs and VLMs. Improved understanding of these models has the potential to significantly benefit sectors requiring robust spatial understanding and navigation. This could lead to more reliable and efficient systems that enhance safety and user experience. As for potential negative societal impacts, our work primarily involves empirical evaluations using synthetic datasets designed to probe spatial reasoning. Therefore, we do not anticipate any direct negative societal impacts arising from our current research.

## C  Experimental Details

### C.1  Software and Hardware

Our experiments are conducted on NVIDIA A100 GPUs. Our implementation is based on Python 3.10 and PyTorch 2.1.2.

## C.2 Hyperparameters and Error Bars

The hyperparameters discussed in this paper pertain to the decoding strategies of each model. For deterministic decoding, we adhere to the default settings specified for each model. The models employing deterministic decoding (argmax) include Bunny-Phi-2-SigLIP, CogAgent, CogVLM, InstructBLIP-Vicuna-13B, and InstructBLIP-Vicuna-7B. For non-deterministic decoding, we utilize the default hyperparameters provided by Hugging Face (for instance, Top-P is set at $0.9$ and the temperature at $0.2$ for LLaVA-1.6). Error bars for the main results (Figure 5) are obtained with three independent runs.

## C.3 Model Checkpoints

For most open-source models, we use the checkpoints provided by Hugging Face as shown in Table 3. For Bunny variants (Bunny-Phi-1.5-EVA, Bunny-Phi-1.5-SigLIP and Bunny-Phi-2-EVA), we use merged weights following instructions in `https://github.com/BAAI-DCAI/Bunny/`.

| Model Name | Link |
|---|---|
| LLaMA-2-13B | https://huggingface.co/meta-llama/Llama-2-13b-chat-hf |
| LLaMA-2-7B | https://huggingface.co/meta-llama/Llama-2-7b-chat-hf |
| LLaMA-3-8B | https://huggingface.co/meta-llama/Meta-Llama-3-8B-Instruct |
| Mistral-7B | https://huggingface.co/mistralai/Mistral-7B-Instruct-v0.2 |
| Nous-Hermes-2-Yi-34B | https://huggingface.co/NousResearch/Nous-Hermes-2-Yi-34B |
| Phi-2 | https://huggingface.co/microsoft/phi-2 |
| Vicuna-13B-1.5 | https://huggingface.co/lmsys/vicuna-13b-v1.5 |
| Vicuna-7B-1.5 | https://huggingface.co/lmsys/vicuna-7b-v1.5 |
| LLaVA-1.6-34B | https://huggingface.co/liuhaotian/llava-v1.6-34b |
| LLaVA-1.6-Mistral-7B | https://huggingface.co/liuhaotian/llava-v1.6-mistral-7b |
| LLaVA-1.6-Vicuna-13B | https://huggingface.co/liuhaotian/llava-v1.6-vicuna-13b |
| LLaVA-1.6-Vicuna-7B | https://huggingface.co/liuhaotian/llava-v1.6-vicuna-7b |
| Bunny-Phi-2-SigLIP | https://huggingface.co/BAAI/Bunny-v1_0-3B |
| Qwen-VL | https://huggingface.co/Qwen/Qwen-VL-Chat |
| CogAgent | https://huggingface.co/THUDM/cogagent-vqa-hf |
| CogVLM | https://huggingface.co/THUDM/cogvlm-chat-hf |
| InstructBLIP-Vicuna-13B | https://huggingface.co/Salesforce/instructblip-vicuna-13b |
| InstructBLIP-Vicuna-7B | https://huggingface.co/Salesforce/instructblip-vicuna-7b |

Table 3: Model checkpoints from Hugging Face.

## C.4 Detailed Illustration of Datasets

In the main text, we abbreviated the textual descriptions in Figure 1, Figure 2 due to space constraints. In this section, we provide the full descriptions for each task to facilitate a better understanding of spatial reasoning tasks. The complete illustrations are displayed in Figure 14 for `Spatial-Map` and Figure 15 for `Maze-Nav`, respectively. Three questions (termed Q1 to Q3) are associated with each sample in tasks `Spaitial-Map`, `Maze-Nav`, and `Spatial-Grid`.

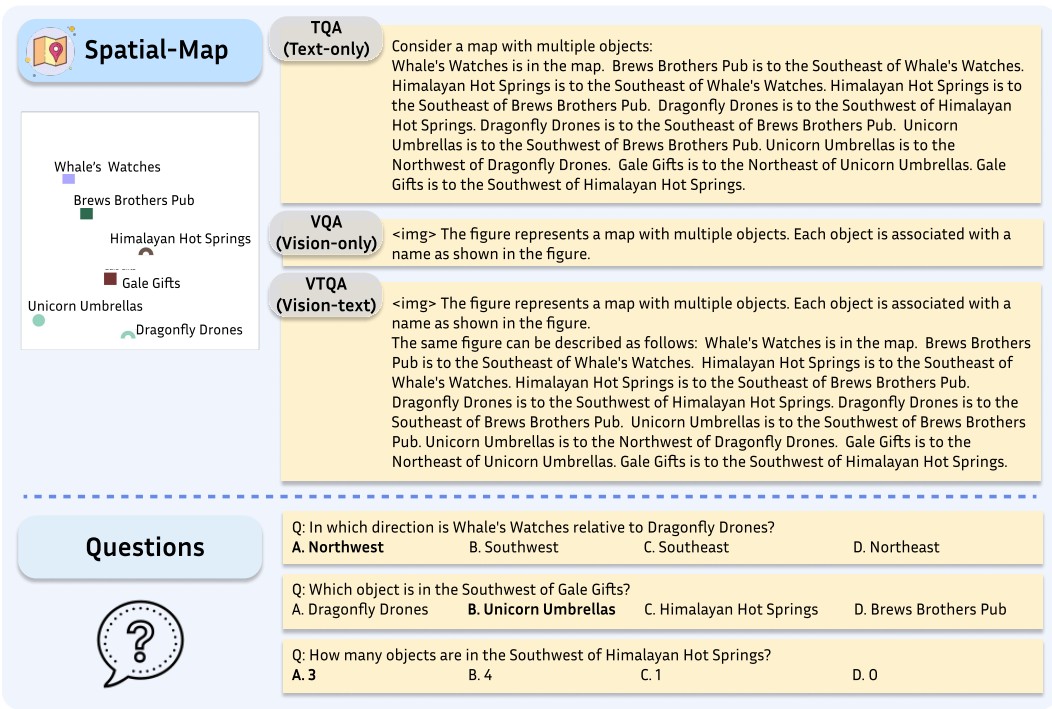

Figure 14: Illustration of the `Spaitial-Map` task with complete textual descriptions in TQA and VTQA.

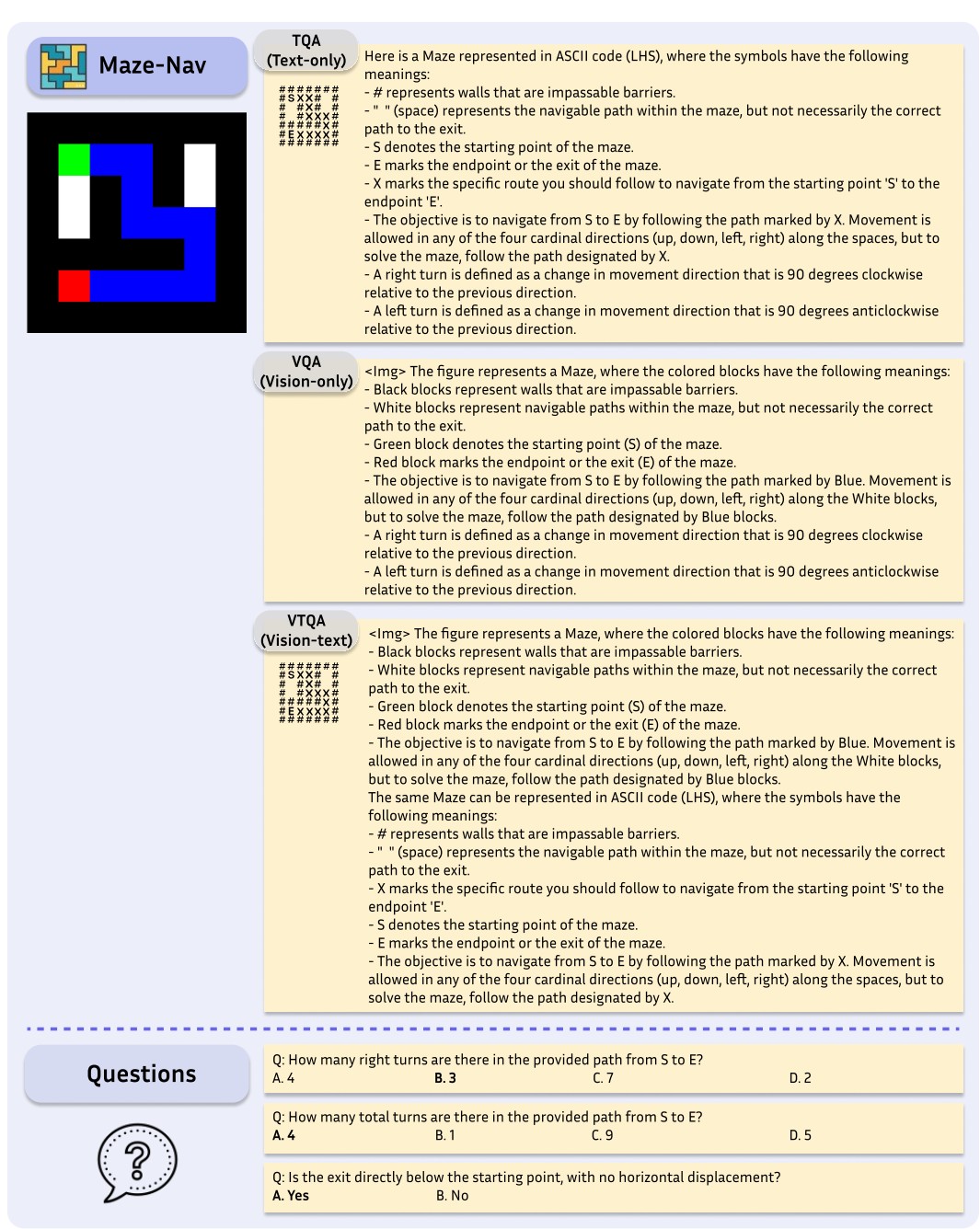

**Maze-Nav**

**TQA (Text-only)**

```
#######
#SXX# #
# #X# #
# #XXX#
#####X#
#EXXXX#
#######
```

Here is a Maze represented in ASCII code (LHS), where the symbols have the following meanings:
- # represents walls that are impassable barriers.
- " " (space) represents the navigable path within the maze, but not necessarily the correct path to the exit.
- S denotes the starting point of the maze.
- E marks the endpoint or the exit of the maze.
- X marks the specific route you should follow to navigate from the starting point 'S' to the endpoint 'E'.
- The objective is to navigate from S to E by following the path marked by X. Movement is allowed in any of the four cardinal directions (up, down, left, right) along the spaces, but to solve the maze, follow the path designated by X.
- A right turn is defined as a change in movement direction that is 90 degrees clockwise relative to the previous direction.
- A left turn is defined as a change in movement direction that is 90 degrees anticlockwise relative to the previous direction.

**VQA (Vision-only)**

 The figure represents a Maze, where the colored blocks have the following meanings:
- Black blocks represent walls that are impassable barriers.
- White blocks represent navigable paths within the maze, but not necessarily the correct path to the exit.
- Green block denotes the starting point (S) of the maze.
- Red block marks the endpoint or the exit (E) of the maze.
- The objective is to navigate from S to E by following the path marked by Blue. Movement is allowed in any of the four cardinal directions (up, down, left, right) along the White blocks, but to solve the maze, follow the path designated by Blue blocks.
- A right turn is defined as a change in movement direction that is 90 degrees clockwise relative to the previous direction.
- A left turn is defined as a change in movement direction that is 90 degrees anticlockwise relative to the previous direction.

**VTQA (Vision-text)**

```
#######
#SXX# #
# #X# #
# #XXX#
#####X#
#EXXXX#
#######
```

 The figure represents a Maze, where the colored blocks have the following meanings:
- Black blocks represent walls that are impassable barriers.
- White blocks represent navigable paths within the maze, but not necessarily the correct path to the exit.
- Green block denotes the starting point (S) of the maze.
- Red block marks the endpoint or the exit (E) of the maze.
- The objective is to navigate from S to E by following the path marked by Blue. Movement is allowed in any of the four cardinal directions (up, down, left, right) along the White blocks, but to solve the maze, follow the path designated by Blue blocks.
The same Maze can be represented in ASCII code (LHS), where the symbols have the following meanings:
- # represents walls that are impassable barriers.
- " " (space) represents the navigable path within the maze, but not necessarily the correct path to the exit.
- X marks the specific route you should follow to navigate from the starting point 'S' to the endpoint 'E'.
- S denotes the starting point of the maze.
- E marks the endpoint or the exit of the maze.
- The objective is to navigate from S to E by following the path marked by X. Movement is allowed in any of the four cardinal directions (up, down, left, right) along the spaces, but to solve the maze, follow the path designated by X.

**Questions**

Q: How many right turns are there in the provided path from S to E?
A. 4          **B. 3**          C. 7          D. 2

Q: How many total turns are there in the provided path from S to E?
**A. 4**          B. 1          C. 9          D. 5

Q: Is the exit directly below the starting point, with no horizontal displacement?
**A. Yes**          B. No

Figure 15: Illustration of the `Maze-Nav` task with complete textual descriptions in TQA, VQA, and VTQA.

# D   Further Ablation Studies and Discussions

**TQA vs. VTQA: No definitive winner.**   In Section 3, we demonstrated the advantages of text-only input (TQA) based on LLMs over vision-only input (VQA) based on VLMs. Subsequently, given the same VLM, the comparison between VTQA and VQA in Section 4 reveals that VLMs rely less on visual information when sufficient textual clues are provided. Curious readers seeking a quantitative comparison between TQA and VTQA can refer to the results illustrated in Figure 16. Unlike the other comparisons, no definitive winner emerges between TQA and VTQA across all spatial reasoning tasks, model sizes, and architectures. For instance, the VTQA performance of InstructBLIP-Vicuna-13B surpasses Vicuna-13B in the `Maze-Nav` task, whereas Vicuna-13B outperforms InstructBLIP-Vicuna-13B in the `Spatial-Map` task. This indicates that text-only inputs (with language models) still hold an advantage in certain spatial reasoning tasks.

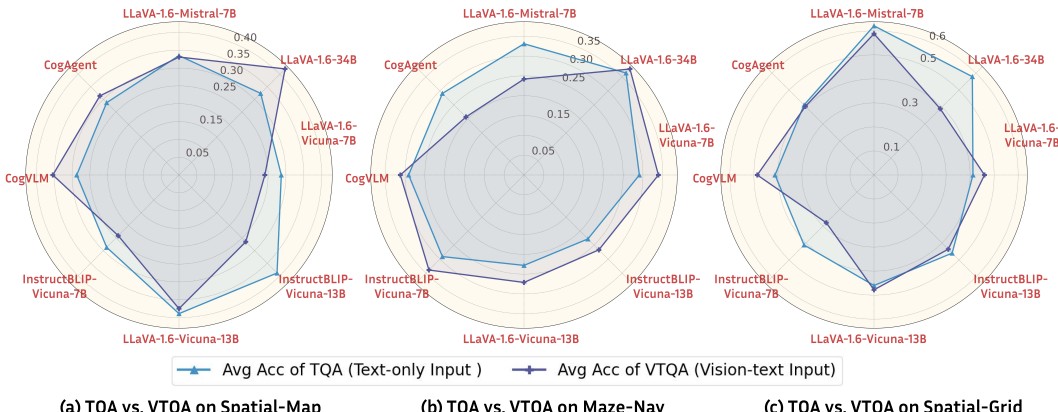

Figure 16: TQA (LLM) vs. VTQA on spatial reasoning tasks. Each vertex on the spider plot represents the Avg Acc of a (LLM, VLM) pair with the same language model backbone. For LLM, we use the Text-only input (*i.e.*, TQA); for VLM, we use Vision-text input (*i.e.*, VTQA).

**Impact of prompting techniques.**   In line with our approach to sampling strategies, our primary goal in choosing prompting techniques is to report the best model performance given the same question. The prompt technique we use, which asks for step-by-step explanation is the most effective query among others in our initial studies. As a concrete example, we compare the original prompting strategy, "First, provide a concise answer in one sentence. Then, elaborate on the reasoning behind your answer in a detailed, step-by-step explanation" (step-by-step explanation), with a simpler prompt, Answer: (completion). Results in Table 4 show that the simpler completion prompt consistently underperforms compared to the step-by-step explanation prompt.

| Input Modality | Model | Avg Acc (completion) | Avg Acc (step-by-step explanation) |
|---|---|---|---|
| Text-only | Mistral-7B | 0.61 | 0.62 |
| | Vicuna-13B-1.5 | 0.25 | 0.46 |
| | Vicuna-7B-1.5 | 0.31 | 0.41 |
| Vision-only | LLaVA-1.6-Mistral-7B | 0.47 | 0.47 |
| | LLaVA-1.6-Vicuna-13B | 0.25 | 0.40 |
| | LLaVA-1.6-Vicuna-7B | 0.26 | 0.31 |
| Vision-text | LLaVA-1.6-Mistral-7B | 0.59 | 0.59 |
| | LLaVA-1.6-Vicuna-13B | 0.26 | 0.48 |
| | LLaVA-1.6-Vicuna-7B | 0.30 | 0.46 |

Table 4: Results for completion and step-by-step prompts.

**Impact of temperature for decoding strategies.**   We conducted an ablation study to examine how different temperatures affect the model performance. A higher temperature allows for more diversity in model responses. Most models consistently underperform when the temperature is set to 1.0 compared to 0.2 (our default) as shown in Table 5.

| Input Modality | Model | Avg Acc (temperature=1) | Avg Acc (temperature=0.2) |
|---|---|---|---|
| Text-only | Mistral-7B | 0.62 | 0.62 |
| | Vicuna-13B-1.5 | 0.37 | 0.46 |
| | Vicuna-7B-1.5 | 0.36 | 0.41 |
| Vision-only | LLaVA-1.6-Mistral-7B | 0.37 | 0.47 |
| | LLaVA-1.6-Vicuna-13B | 0.32 | 0.40 |
| | LLaVA-1.6-Vicuna-7B | 0.26 | 0.31 |
| Vision-text | LLaVA-1.6-Mistral-7B | 0.46 | 0.59 |
| | LLaVA-1.6-Vicuna-13B | 0.37 | 0.48 |
| | LLaVA-1.6-Vicuna-7B | 0.33 | 0.46 |

Table 5: Results by varying different temperatures for decoding strategies.

# E Results on Spatial-Real task

Table 6 presents the performance on the `Spatial-Real` task. The same trends observed in synthetic tasks persist with real images (see VQA vs. VTQA, TQA (LLM) vs. VTQA, TQA (LLM) vs. VQA in Table 2). Notably, compared to synthetic tasks (Figure 5 and Figure 11), the overall accuracy on `Spatial-Real` is increased across all three input types (TQA, VQA, and VTQA). However, the modality gap (accuracy difference between VTQA and VQA) widens significantly, from 7.0% on synthetic tasks to 30.0% on `Spatial-Real`. This indicates that the performance disparity is more pronounced on natural images.

| Input Format | Model | Average Accuracy |
|---|---|---|
| Text-only (TQA (LLM)) | Vicuna-13B-1.5 | 0.845 |
| | Vicuna-7B-1.5 | 0.716 |
| | Mistral-7B | 0.800 |
| | LLaMA-3-8B | 0.884 |
| Vision-only (VQA) | CogVLM | 0.419 |
| | Qwen-VL | 0.329 |
| | LLaVA-1.6-Mistral-7B | 0.368 |
| | CogAgent | 0.400 |
| | LLaVA-1.6-Vicuna-7B | 0.432 |
| | LLaVA-1.6-Vicuna-13B | 0.445 |
| Vision-text (VTQA) | CogVLM | 0.594 |
| | Qwen-VL | 0.729 |
| | LLaVA-1.6-Mistral-7B | 0.761 |
| | CogAgent | 0.471 |
| | LLaVA-1.6-Vicuna-13B | 0.832 |
| | LLaVA-1.6-Vicuna-7B | 0.806 |

Table 6: Performance on the `Spatial-Real` task. The same trends still hold on real images: VQA vs. VTQA, TQA (LLM) vs. VTQA, and TQA (LLM) vs. VQA.

# F Detailed Experimental Results

Results for proprietary models are summarized in Table 7. In Section 3 and Section 4, to clearly illustrate the impact of modality, we present the results averaged over all questions in Figure 6 for VQA vs. TQA and Figure 11 for VQA vs. VTQA. This section provides a comprehensive breakdown of results for individual questions. Detailed comparative results for `Spatial-Map`, `Maze-Nav`, and `Spatial-Grid` are shown in Table 8, Table 9, and Table 10, respectively. We compare LLM and VLM with the same language model backbone. For the VLM assessments, we consider inputs in both vision-only (VQA) and vision-text (VTQA) formats.

| Input Format | Model Name | Spatial-Map Acc | Maze-Nav Acc | Spatial-Grid Acc |
|---|---|---|---|---|
| Text-only | Claude 3 Opus | 0.619 | 0.277 | 0.957 |
| | Gemini Pro 1.0 | 0.453 | 0.223 | 0.620 |
| | GPT-4o | 0.616 | 0.353 | 0.986 |
| | GPT-4V | 0.555 | 0.321 | 0.923 |
| | GPT-4 | 0.504 | 0.353 | 0.939 |
| Vision-only | Claude 3 Opus | 0.348 | 0.163 | 0.455 |
| | Gemini Pro 1.0 | 0.263 | 0.219 | 0.512 |
| | GPT-4o | 0.617 | 0.410 | 0.833 |
| | GPT-4V | 0.237 | 0.263 | 0.668 |
| Vision-text | Claude 3 Opus | 0.606 | 0.332 | 0.888 |
| | Gemini Pro 1.0 | 0.456 | 0.203 | 0.687 |
| | GPT-4o | 0.643 | 0.382 | 0.989 |
| | GPT-4V | 0.490 | 0.333 | 0.924 |

Table 7: Detailed results for proprietary models.

| Input Format | Model Name | Q1 Acc | Q2 Acc | Q3 Acc | Avg Acc |
|---|---|---|---|---|---|
| Text-only | Mistral-7B | 0.40 | 0.27 | 0.34 | 0.34 |
| | Nous-Hermes-2-Yi-34B | 0.45 | 0.22 | 0.30 | 0.32 |
| | Vicuna-7B-1.5 | 0.39 | 0.40 | 0.07 | 0.29 |
| | Vicuna-13B-1.5 | 0.44 | 0.52 | 0.21 | 0.39 |
| Vision-only | LLaVA-1.6-Mistral-7B | 0.26 | 0.37 | 0.13 | 0.25 |
| | LLaVA-1.6-34B | 0.30 | 0.28 | 0.30 | 0.29 |
| | LLaVA-1.6-Vicuna-7B | 0.27 | 0.32 | 0.26 | 0.28 |
| | InstructBLIP-Vicuna-13B | 0.22 | 0.21 | 0.08 | 0.17 |
| | LLaVA-1.6-Vicuna-13B | 0.31 | 0.32 | 0.15 | 0.26 |
| | InstructBLIP-Vicuna-7B | 0.25 | 0.18 | 0.06 | 0.16 |
| | CogVLM | 0.24 | 0.39 | 0.13 | 0.25 |
| | CogAgent | 0.29 | 0.34 | 0.10 | 0.25 |
| Vision-text | LLaVA-1.6-Mistral-7B | 0.43 | 0.42 | 0.15 | 0.33 |
| | LLaVA-1.6-34B | 0.58 | 0.47 | 0.21 | 0.42 |
| | LLaVA-1.6-Vicuna-7B | 0.35 | 0.25 | 0.12 | 0.24 |
| | InstructBLIP-Vicuna-13B | 0.39 | 0.33 | 0.08 | 0.27 |
| | LLaVA-1.6-Vicuna-13B | 0.49 | 0.49 | 0.14 | 0.38 |
| | InstructBLIP-Vicuna-7B | 0.35 | 0.25 | 0.12 | 0.24 |
| | CogVLM | 0.40 | 0.52 | 0.14 | 0.35 |
| | CogAgent | 0.37 | 0.45 | 0.12 | 0.31 |

Table 8: Detailed results for the `Spatial-Map` task. We compare LLM and VLM with the same language model backbone. For VLMs, we consider both vision-only and vision-text input format. The averaged results are summarized in Figure 6 and Figure 11.

| Input Format | Model Name | Q1 Acc | Q2 Acc | Q3 Acc | Avg Acc |
|---|---|---|---|---|---|
| Text-only | Mistral-7B | 0.43 | 0.23 | 0.33 | 0.33 |
| | Nous-Hermes-2-Yi-34B | 0.25 | 0.18 | 0.67 | 0.36 |
| | Vicuna-7B-1.5 | 0.11 | 0.13 | 0.63 | 0.29 |
| | Vicuna-13B-1.5 | 0.20 | 0.15 | 0.33 | 0.23 |
| Vision-only | LLaVA-1.6-Mistral-7B | 0.25 | 0.10 | 0.34 | 0.23 |
| | LLaVA-1.6-34B | 0.27 | 0.23 | 0.67 | 0.39 |
| | LLaVA-1.6-Vicuna-7B | 0.17 | 0.22 | 0.65 | 0.35 |
| | InstructBLIP-Vicuna-13B | 0.15 | 0.09 | 0.67 | 0.31 |
| | LLaVA-1.6-Vicuna-13B | 0.33 | 0.20 | 0.33 | 0.29 |
| | InstructBLIP-Vicuna-7B | 0.21 | 0.28 | 0.33 | 0.27 |
| | CogVLM | 0.13 | 0.16 | 0.66 | 0.32 |
| | CogAgent | 0.16 | 0.22 | 0.33 | 0.24 |
| Vision-text | LLaVA-1.6-Mistral-7B | 0.27 | 0.10 | 0.36 | 0.24 |
| | LLaVA-1.6-34B | 0.28 | 0.18 | 0.68 | 0.38 |
| | LLaVA-1.6-Vicuna-7B | 0.16 | 0.23 | 0.63 | 0.34 |
| | InstructBLIP-Vicuna-13B | 0.11 | 0.11 | 0.58 | 0.27 |
| | LLaVA-1.6-Vicuna-13B | 0.31 | 0.18 | 0.33 | 0.27 |
| | InstructBLIP-Vicuna-7B | 0.16 | 0.23 | 0.63 | 0.34 |
| | CogVLM | 0.14 | 0.20 | 0.60 | 0.31 |
| | CogAgent | 0.14 | 0.14 | 0.34 | 0.21 |

Table 9: Detailed results for the `Maze-Nav` task. We compare LLM and VLM with the same language model backbone. For VLMs, we consider both vision-only and vision-text input format. The averaged results are summarized in Figure 6 and Figure 11.

| Input Format | Model Name | Q1 Acc | Q2 Acc | Q3 Acc | Avg Acc |
|---|---|---|---|---|---|
| Text-only | Mistral-7B | 0.47 | 0.73 | 0.66 | 0.62 |
| | Nous-Hermes-2-Yi-34B | 0.33 | 0.91 | 0.50 | 0.58 |
| | Vicuna-7B-1.5 | 0.38 | 0.48 | 0.38 | 0.41 |
| | Vicuna-13B-1.5 | 0.34 | 0.65 | 0.39 | 0.46 |
| Vision-only | LLaVA-1.6-Mistral-7B | 0.28 | 0.69 | 0.45 | 0.47 |
| | LLaVA-1.6-34B | 0.32 | 0.29 | 0.21 | 0.27 |
| | LLaVA-1.6-Vicuna-7B | 0.30 | 0.38 | 0.26 | 0.31 |
| | InstructBLIP-Vicuna-13B | 0.24 | 0.24 | 0.21 | 0.23 |
| | LLaVA-1.6-Vicuna-13B | 0.30 | 0.58 | 0.30 | 0.40 |
| | InstructBLIP-Vicuna-7B | 0.27 | 0.25 | 0.24 | 0.25 |
| | CogVLM | 0.25 | 0.38 | 0.34 | 0.32 |
| | CogAgent | 0.33 | 0.33 | 0.29 | 0.32 |
| Vision-text | LLaVA-1.6-Mistral-7B | 0.32 | 0.79 | 0.65 | 0.59 |
| | LLaVA-1.6-34B | 0.44 | 0.40 | 0.33 | 0.39 |
| | LLaVA-1.6-Vicuna-7B | 0.36 | 0.68 | 0.34 | 0.46 |
| | InstructBLIP-Vicuna-13B | 0.33 | 0.59 | 0.39 | 0.44 |
| | LLaVA-1.6-Vicuna-13B | 0.29 | 0.78 | 0.36 | 0.48 |
| | InstructBLIP-Vicuna-7B | 0.28 | 0.29 | 0.27 | 0.28 |
| | CogVLM | 0.30 | 0.82 | 0.35 | 0.49 |
| | CogAgent | 0.33 | 0.52 | 0.36 | 0.40 |

Table 10: Detailed results for the `Spatial-Grid` task. We compare LLM and VLM with the same language model backbone. The averaged results are summarized in Figure 6 and Figure 11.

