# OpenReview forum: "Is A Picture Worth A Thousand Words? Delving Into Spatial Reasoning for Vision Language Models"
_NeurIPS.cc/2024/Conference — NeurIPS 2024 poster_

### Official Review · Reviewer_WNLP · 2024-07-08

**Soundness:** 3
**Presentation:** 3
**Contribution:** 3
**Rating:** 6
**Confidence:** 4

**Summary:**

This work presents 3 new synthetically generated VQA evaluation benchmarks and conducts a comprehensive evaluation of the limitations of current  VLMs in spatial reasoning tasks. These tasks include spatial relationships, navigation, position understanding, and counting. The authors demonstrate using their introduced benchmarks that VLMs have limited performance on tasks requiring detailed visual understanding and reasoning of images.

**Strengths:**

Evaluating the spatial reasoning capabilities of SoTA VLMs is highly impactful and very relevant. I find this work novel, intriguing, and highly useful as it points to very important limitations in VLMs.  The experimental analysis is very comprehensive, incorporating both several open-source VLMs and proprietary models. Additionally, the paper is well-written and structured, contributing to its overall readability and clarity.

**Weaknesses:**

Spatial Understanding and Evaluation: The use of synthetic data in the evaluation, while valuable, may introduce confounding factors unrelated to the task of spatial understanding/reasoning.
* For instance, a vision-only model with limited OCR capabilities may perform poorly on the spatial-map task, regardless of its actual spatial reasoning ability. It is challenging to disentangle the contributions of OCR performance from spatial understanding in such benchmark.
* The synthetic maze-navigation and spatial-grid images might be out-of-distribution for some models, especially open-source ones. This could also explain why a noise image improves the accuracy of Llava-1.6 on maze-navigation tasks, while a similarly out-of-distribution maze image does not necessarily harm or consistently improve the  performance. It is important to consider the potential impact of out-of-distribution data on open-source models. By looking at fig 12 it looks the performance of the latest gpt4o on vision-only is similar to vision-and-text. That said, I still believe the work remains valuable and important.

Human Performance Comparison:
* The claim on line 172 that "the performance of these models still lags significantly behind human levels" would benefit from concrete numbers to substantiate it. Providing specific human performance metrics on these tasks or citations here could would make the claim more robust. Alternatively, softening the statement to reflect the need for further comparison could be considered.

Sampling Strategies and Prompting Techniques:
* Exploring different sampling strategies or reasoning/prompting techniques could yield valuable insights into the model's performance. It would be beneficial to include discussions on how these variations impact the results. To my understanding only a simple prompting technique is utilized "step-by-step explanation". Was there any particular reason to append the selected prompt vs others.
In line 157, the paper also mentioned "For each model, we adopt the default configurations and decoding strategies, e.g., argmax for deterministic decoding and top-p for non-deterministic decoding. " It would be very useful to be more specific, e.g. what top-p and temperature was used.  How "deterministic" was the deterministic decoding in API based models?

VQA Evaluation Benchmark Details:
* The paper would benefit from clearly specifying the size of each VQA evaluation benchmark in terms of the number of samples or data points. This information is crucial for understanding the scope and scale of the evaluations conducted. I might have missed it but I couldn't find it in the paper. I would encourage the authors to particular address this in rebuttals.

**Questions:**

please see above

**Limitations:**

Yes. It is discussed and properly addressed.

---

> ### Author Rebuttal · Authors · 2024-08-07
>
> We are grateful for your support of our work and insightful comments!
>
>
> > *Q1: The use of synthetic data in the evaluation, while valuable, may introduce confounding factors (e.g., (1) It is challenging to disentangle the contributions of OCR)*. (2) The synthetic data might be out-of-distribution for some models. It is important to consider the potential impact of out-of-distribution data...That said, I still believe the work remains valuable and important.
>
> Thank you for your insightful feedback, and we appreciate your recognition of the value and importance of our work! We agree that disentangling OCR capabilities from spatial reasoning abilities in VQA (Vision-only) input format for VLMs presents a significant challenge. This challenge motivated our development of the VTQA (Vision-Text) input format. In VTQA, we aim to enhance VLMs' OCR capabilities by providing a detailed textual description of the objects. This approach helps to mitigate the impact of OCR abilities and emphasizes spatial reasoning. Additionally, our benchmark avoids questions with direct textual answers provided in the prompt, requiring the model to engage in genuine reasoning to arrive at the correct answer.
>
> Indeed, the synthetic datasets might be out-of-distribution for some models. We have intentionally include such synthetic examples in our benchmarks to test the robustness and generalization ability. By doing so, we ensure that good performance is not merely a result of data memorization during web-scale pre-training. We hope our work will provide a foundation for future advancements in VLM development.
>
> We further curate a real-world task, Spatial-Real, based on real images with dense captions [1]. Detailed results are shown in **Table 1** in the one-page PDF. We find that the trends (summarized comparisons VQA vs. VTQA, TQA (LLM) vs. VTQA, TQA (LLM) vs. VQA in **Table G.2**) stated in the paper still holds. The modality gap (accuracy difference between VTQA and VQA) even grows from 7.0% on synthetic benchmarks to 30.0% on Spatial-Real on average.
>
> > *Q2: The claim on line 172 that "the performance of these models still lags significantly behind human levels" would benefit from concrete numbers to substantiate it.*
>
> Thank you for your suggestion! We have conducted a human evaluation on a subset of 900 samples for the three tasks. This subset revealed an average human accuracy rate exceeding 96% (VQA). While these results suggest high human performance on the evaluated tasks, we recognize the limitation of not extending this comparison across all 13,500 data points due to budget constraint. We are committed to an extensive human evaluation on the full dataset and will include the results in the revised manuscript.
>
> > *Q3: The specifics of decoding configs and the impact of prompting techniques.*
>
> Thanks for the suggestion! We provide details for each below:
>
> **Decoding configurations:** We have included detailed decoding configurations in Appendix B.2. Our primary approach is to use the default decoding strategies provided by each model to ensure fair comparisons and report the best performance for each model given the same input. Specifically, we use deterministic decoding (argmax) for the following models: Bunny-Phi-2-SigLIP, CogAgent, CogVLM, InstructBLIP-Vicuna-13B, and InstructBLIP-Vicuna-7B. For all other models, we employ non-deterministic decoding with Top-p set to 0.9 and temperature set to 0.2.
>
> **Impact of Temperature**: We conducted an ablation study to examine how different temperatures affect the model performance. A higher temperature allows for more diversity in model responses. Most models consistently underperform when the temperature is set to 1.0 compared to 0.2 (our default), as shown below. We have included the complete results of this study in Appendix E.
>
> |Input modality|Model|Avg Acc (temperature=1)|Avg Acc (temperature=0.2)|
> |-|-|-|-|
> |Text-only|Mistral-7B| 0.62|0.62|
> |Text-only|Vicuna-13B-1.5| 0.37|0.46|
> |Text-only|Vicuna-7B-1.5| 0.36|0.41|
> |Vision-only|LLaVA-1.6-Mistral-7B|0.37|0.47|
> |Vision-only|LLaVA-1.6-Vicuna-13B|0.32|0.40|
> |Vision-only|LLaVA-1.6-Vicuna-7B|0.26|0.31|
> |Vision-text|LLaVA-1.6-Mistral-7B|0.46|0.59|
> |Vision-text|LLaVA-1.6-Vicuna-13B|0.37|0.48|
> |Vision-text|LLaVA-1.6-Vicuna-7B|0.33|0.46|
>
> **Prompting techniques:** In line with our approach to sampling strategies, our primary goal in choosing prompting techniques is to report the best model performance given the same question. The prompt technique we use, which asks for step-by-step explanation is the most effective query among others in our initial studies.
>
> As a concrete example, we compare the original prompting strategy, *"First, provide a concise answer in one sentence. Then, elaborate on the reasoning behind your answer in a detailed, step-by-step explanation"* (step-by-step explanation), with a simpler prompt, *Answer:* (completion).
>
> The results are shown below. We can see that the simpler completion prompt consistently underperforms compared to the step-by-step explanation prompt.
>
> |Input Modality|Model|Avg Acc (completion)|Avg Acc (step-by-step explanation)|
> |-|-|-|-|
> |Text-only|Mistral-7B|0.61|0.62|
> |Text-only|Vicuna-13B-1.5|0.25|0.46|
> |Text-only|Vicuna-7B-1.5| 0.31| 0.41|
> |Vision-only|LLaVA-1.6-Mistral-7B|0.47|0.47|
> |Vision-only|LLaVA-1.6-Vicuna-13B|0.25|0.40|
> |Vision-only|LLaVA-1.6-Vicuna-7B| 0.26|0.31|
> |Vision-text|LLaVA-1.6-Mistral-7B|0.59|0.59|
> |Vision-text|LLaVA-1.6-Vicuna-13B|0.26|0.48|
> |Vision-text|LLaVA-1.6-Vicuna-7B|0.30|0.46|
>
> > *Q4: sample size of each VQA benchmark*
>
> Thank you for pointing this out! In our benchmark, each task (Spatial-Map, Maze-Nav, Spatial-Grid) contains 4,500 samples, resulting in a total of 13,500 samples. Additionally, our benchmarks are designed to be easily scalable. We have included this information in the revised manuscript (Appendix B).
>
> [1] Urbanek et al., A Picture is Worth More Than 77 Text Tokens: Evaluating CLIP-Style Models on Dense Captions, CVPR 2024.

---

> > ### Comment · Reviewer_WNLP · 2024-08-12
> > **Thanks for the response**
> >
> > I thank the authors for the very comprehensive  rebuttal and new experiments added to the papers.
> > I have also read and appreciate other reviewers comments and the authors response. I think this is an important paper, technically solid with potential moderate-to-high impact. I keep my score of 6, and recommend acceptance.

---

> > > ### Author Response · Authors · 2024-08-12
> > >
> > > Dear Reviewer WNLP,
> > >
> > > Thank you for your time and thoughtful feedback on our paper. We greatly appreciate your positive recommendation!
> > >
> > > Best,
> > > Authors

---

### Official Review · Reviewer_JHoY · 2024-07-13

**Soundness:** 2
**Presentation:** 3
**Contribution:** 3
**Rating:** 6
**Confidence:** 4

**Summary:**

The authors make a benchmark to test the spatial reasoning of multimodal language models (MLMs) using synthetic data consisting mostly of diagrams, mazes, and charts. They benchmark a lot of existing MLMs. Their key findings are: 1) Spatial reasoning is limited in most MLMs. 2) MLMs rely heavily on textual information and not the visual, and 3) interestingly, the VLMs are better at utilizing spatial text information than LLMs suggesting multimodal training helps understand the visual natural of language even though VLMs tend to rely less on visual information during test time.

**Strengths:**

The idea of stress testing MLMs with spatial and figure understanding has been a recent trend and is very useful to push these models to do real-world tasks.
The analysis done is interesting and provides some useful insights into how these models use visual or textual information.

**Weaknesses:**

- Some of the claims may be too broad to make based on results of just this dataset. For instance, the models using more textual information than visual. This could simply be because the visual inputs in this benchmark are of a starkly different domain than what the models are trained on (real images instead of synthetic graphs and mazes and charts), whereas the domain gap in language is much lesser since the language tokens remain the same. Hence, we can only make the statement that the model is using more textual cues for such kinds of data. It is unclear if that is also the case on real image data.

- There are many ways to test spatial reasoning using real VQA style datasets like BLINK, MMBench etc. So , it is unclear why they came up with these complicated tasks. Is there any real-world relevance for such tasks?

- The authors mention they opt for synthetic data due to controllability, but all the analysis they do can be simply done on real image QA datasets like GQA.  For instance, for checking vision vs textual reliance, just feed in scene graphs vs the image to an MLM from GQA. Or, for mismatched image and text, randomly pair image and QA. At least a discussion on why such a benchmark is needed is missing. What special analysis in the paper is enabled by this benchmark that couldn't be enabled by an already existing benchmark like Visual Genome, GQA, BLINK etc.

- Some settings could be explained more clearly in the paper. For instance, for the vision-only input - I am guessing the authors mean that they do not describe the image in text, but the question is still in text, correct?

- A nit pick, but some grammar could be improved. e.g., intro, the first line (line 18):" had a transformation effect on research" - a transformational effect not affect. Other grammatical errors are also occasionally present in the paper.

-  Another nit pick, but this paper seems more fitting for the datasets and benchmarks track.

**Questions:**

See above concerns.

Overall I like the direction of the paper. But some analysis justifying the need of such type of synthetic data (mazes and custom diagrams) instead of real images or actual graphs (useful for scientific figures) would be nice.

**Limitations:**

The authors discuss some limitations of models, but not the benchmark. What can this benchmark power, and what are some analyses that the benchmark currently cannot power that would be good to have?

---

> ### Author Rebuttal · Authors · 2024-08-07
>
> We sincerely appreciate your feedback and comments!
>
> > *Q1: Some claims may be too broad; we can only state the model uses more textual cues for such data*
>
> Thanks for the comments and the suggestion! We acknowledge the visual domain gap and have added remarks in our revised manuscript. We intentionally include such synthetic data in our benchmarks to avoids potential data leakage from popular visual benchmarks, ensuring performance isn't due to data memorization during pre-training. Additionally, our benchmarks shift the focus from object recognition to evaluating spatial reasoning abilities with numerous objects. Similar paradigms have been explored in recent works evaluating visual diagram understanding for mathematical reasoning such as [1].
>
> > *Q2: Why need such a benchmark is missing. What analysis does it enable that existing benchmarks like Visual Genome, GQA, and BLINK do not?*
>
> We would like to highlight some key rationales of our benchmark:
> 1. Existing benchmarks focus only on VQA, where the image is required but the text description is often omitted or optional. Our study further explores spatial reasoning across different settings: TQA (LLM), TQA (VLM), and VTQA, where images or texts can be optional, thereby broadening the scope of tasks.
> 2. We utilize synthetic data due to its controllability, scalability, and the ability to create highly specific scenarios with flexible, long and detailed captions that are not adequately covered by existing benchmarks such as Visual Genome, GQA, and BLINK. While these datasets are valuable, they do not consistently offer the level of complexity we require, such as numerous objects containing dense visual information along with detailed natural language captions that fully convey the image content.
> 3. The textual descriptions in these datasets are often too brief or directly imply the answers. In our work, we provide dense or detailed captions for each image. As a result, no answers can be easily inferred from a short caption. We also try to isolate object detection capability from spatial reasoning ability by simplifying objects to symbols.
> 4. Although our tasks seem complicated, humans can still solve them with near-perfect accuracy. This indicates that the tasks are within the realm of human cognitive capabilities and are therefore realistic for evaluating advanced AI models.
>
> Given the increasing use of VLMs, it is crucial to have a diverse suite of benchmarks to assess their abilities.
> We also believe these awesome VQA datasets are relevant and have cited in our revised manuscript.
>
> > *Q3: Real-world applications/relevance for such synthetic tasks?*
>
> We would like to highlight some of the real-world applications:
> - Diagram and Document Understanding: An increasingly relevant application for MLLMs is the interpretation of digital documents that include a dense array of visual elements. For businesses and educational sectors, the ability to understand symbols, figures, and diagrams within structured layouts is crucial.
> - Map Understanding and Navigation: Our Spatial-Map and Maze-Nav benchmarks simulate map-like environments scattered with numerous objects, such as hotels and stores, each represented by distinct symbols. These configurations are typical in map apps.
> - Warehouse Operations and Traffic Management: autonomous robots navigate through grid-like storage layouts densely packed with objects, requiring precise spatial understanding for efficient item retrieval and restocking. Similarly, in urban traffic management, systems must accurately identify and count vehicles within structured scenes, such as busy intersections where vehicles are compactly arranged in lanes and rows. These capabilities are critical for the safe deployment of MLLM-based systems.
>
> > Q4: Will the statements still hold on real image data?
>
> Thanks for your comments and suggestions! Inspired by your suggestion to explore natural images, we found a very recent work [2] released a Densely Captioned Images (DCI) dataset, featuring detailed captions with over 1000 words per image. However, this dataset lacks questions. Therefore, we carefully curated multiple-choice questions regarding spatial-reasoning (object counting, relation, and position understanding) and annotated the answers. We name this new dataset as **Spatial-Real**. Due to the rebuttal period's time constraints, we have created 200 TQA-VQA-VTQA pairs.
>
> The evaluation results, shown in **Table 1** in the one-page PDF, indicate that the same trends still hold for real images (see VQA vs. VTQA, TQA (LLM) vs. VTQA, TQA (LLM) vs. VQA in **Table G.2**). In addition, compared to Fig 4 and 10 in the paper, overall accuracy increases across all three input modalities (Text-only, Vision-only, Vision-text) in the Spatial-Real benchmark. However, the modality gap (accuracy difference between VTQA and VQA) grows from 7.0% on synthetic benchmarks (avg) to 30.0% (avg) on Spatial-Real.
>
> To ensure full reproducibility, we will open source this new dataset and extend the number of sample pairs to match Spatial-Map, Spatial-Grid, and Maze-Nav. The full results will be included in our revised manuscript.
>
>
> > *W4: Some settings could be clearer (e.g., the meaning of vision-only input)?*
>
> Thanks for the suggestion! We have clarified this in the revised manuscript. Your understanding is correct. We describe the Text-only, Vision-only, and Vision-text input modalities based on how we feed the image information to the models. Vision-only input means the image is fed directly to the models without textual description, while all questions are presented in text.
>
> > *W5: Grammars and typos*
>
> Thanks for the catch! We have carefully examined and fixed grammatical errors and typos throughout.
>
>
> [1] Zhang et al., MathVerse: Does Your Multi-modal LLM Truly See the Diagrams in Visual Math Problems?
>
> [2] Urbanek et al., A Picture is Worth More Than 77 Text Tokens: Evaluating CLIP-Style Models on Dense Captions, CVPR 2024.

---

> ### Comment · Reviewer_JHoY · 2024-08-12
> **Thanks for the response**
>
> These responses are helpful and the introduction of a split with real images also broadens the scope of the paper.
> While some of the tasks seem contrived, performance on such tasks can be seen as basic cognitive tests, and as the authors point out, they can have real downstream applications, such as navigation given maps, etc. However, evidence that improving on this benchmark will actually improve navigation performance is missing.
>
> Some of the concerns are that the Spatial real example in the rebuttal seems to go back to simple image-based QAs again, and it's unclear how it is "spatial reasoning". It is also unclear how it is different from the QAs in existing datasets that also have spatial, counting, etc QAs.
>
> However, I still lean towards recommending accepting the paper since it will help drive research into the cognitive abilities of multimodal foundation models.

---

> > ### Author Response · Authors · 2024-08-14
> >
> > Dear Reviewer JHoY,
> >
> > Thank you again for your valuable feedback. We sincerely appreciate your support of our work.
> >
> > Q: What is new in Spatial-Real compared to existing image-based datasets that also have QAs on spatial understanding and counting?
> >
> > In Spatial-Real, the form of QAs on spatial reasoning (object counting, relation, and position understanding) is similar to those in existing VQA datasets. However, the novel difference is that our dataset construction allows for a comprehensive study of spatial reasoning QAs across different modality settings—TQA (LLM), TQA (VLM), VQA, and VTQA, where images or texts can be optionally provided. This is largely unexplored in Vision and NLP communities. We aim to bridge this gap by offering a unique benchmark that we hope will be valuable and inspire future research, as new multimodal foundation models emerge that can handle interleaved text and image inputs with longer context lengths.
> >
> >
> > Best,
> >
> > Authors

---

### Official Review · Reviewer_i8Xy · 2024-07-13

**Soundness:** 3
**Presentation:** 3
**Contribution:** 3
**Rating:** 5
**Confidence:** 4

**Summary:**

The paper proposes a set of synthetic tests to compare the spatial understanding in VLMs and LLMs. The tests include Spatial-Map, Maze-Nav, and Spatial-Grid, which all include an image, paired with text that describe the image, and a question. Using the synthetic data, the VLMs and LLMs are studied using different modalities, including VQA, TQA, and VTQA, where the models are asked to answer questions based on only the image, only the text, and both image and text. Findings include that the current VLMs are strongly biased towards texts (blind); VLMs outperform LLMs in TQA. Both open models and proprietary models have been tested.

**Strengths:**

1. The TQA, VQA, VTQA test is a smart way to reveal the modality bias in VLMs and LLMs, clearly showing the blindness of VLMs.
2. Intensive results and analysis are provided.

**Weaknesses:**

1. All datasets contain only 2D synthetic images, which is substantially different from real images. At least the images can be made more realistic using computer graphics, like [4,5].
2. The synthetic nature of the tasks introduces artifacts. Good spatial reasoning ability on this task may not generalize to real ones. The results may be highly dependent on whether the model has been trained on a similar task. For example, the performance of GPT-4O almost doubles that of Gemini in this paper, while the difference of these models on real images is not this significant.
3. The observation that VL models are highly biased towards text, and are “blind” visually, has been studied long in the VQA community, even before LLMs [1,2,3, etc.]. The related literature should be discussed.
[1] Eyes wide shut? exploring the visual shortcomings of multimodal llms
[2] Explicit Bias Discovery in Visual Question Answering Models
[3] Making the V in VQA Matter: Elevating the Role of Image Understanding in Visual Question Answering
[4] Clevr: A diagnostic dataset for compositional language and elementary visual reasoning
[5] Super-clevr: A virtual benchmark to diagnose domain robustness in visual reasoning

**Questions:**

1. How does the findings in this paper differ from previous literatures like [1,2,3]. Are the findings different, or are they discovered in a new way as proposed in this paper?
2. Any discussions about the synthetic-real gap?
3. In Spatial-Map, without text descriptions, it looks like the VLMs may describe spatial using “left/right/up/down” instead of “west/east/north/south” - will this happen, and how is it addressed?

---

> ### Author Rebuttal · Authors · 2024-08-07
>
> Thank you for your positive feedback and insightful comments!
>
> > *W1: All datasets contain only 2D synthetic images, which is substantially different from real images.*
>
> We choose synthetic data due to its controllability, scalability, and the ability to create highly specific scenarios with flexible, long and detailed captions that are not adequately covered by existing VQA benchmarks.
>
> In addition, this approach avoids potential data leakage and ensures that model performance is not merely the result of data memorization during web-scale pre-training. Moreover, our benchmarks shift the focus from object recognition to evaluating spatial reasoning abilities involving numerous objects. Similar paradigms have been explored in recent works evaluating visual diagram understanding [6].
>
>
> > *Q1: The observation that VLMs are highly biased towards text, and are “blind” visually, has been studied long in the VQA community, even before LLMs [1,2,3, etc.]. How does the findings in this paper differ from previous literatures like [1,2,3]?**
>
> Thank you for the suggestion! We find these works intriguing and relevant, and have cited and discussed references [1-5] in our revised manuscript. We would like to highlight several key differences in our approach:
>
> 1. **Task Scope and Focus**: Previous works [1-5] primarily focus on the VQA task where images are required but text descriptions are often omitted or optional. In contrast, our study further explores spatial reasoning across different settings: TQA (LLM), TQA (VLM), and VTQA, where images or texts can be optional, thereby broadening the scope and complexity of tasks.
> 2. **Evaluation**: we primarily focus on multimodal language models (MLLM). We treat these models as generative models which are required to "elaborate on the reasoning behind your answer in a detailed, step-by-step explanation" (L154). In contrast, the evaluation strategy is different in prior works [2-5], where the task is often discriminative with no explicit reasoning. Therefore, it remains unknown if previous observations can be naturally transferred to foundation models pre-trained on web-scale data. Concurrent work [1] also evaluates MLLMs but focuses on visual representations by curating pairs of images adversarial for CLIP-based models. In contrast, our data is unrelated to CLIP features and our questions are not designed to be adversarial. Humans can solve our tasks with near-perfect accuracy. This indicates that the tasks are within the realm of human cognitive capabilities and are realistic for evaluating modern MLLMs.
> 3. **The Textual Representation of Images**: the textual descriptions in prior VQA benchmarks are often too brief or directly imply the answers. In contrast, we provide dense or detailed captions for each image. Therefore, our benchmarks do not include questions where answers can be easily inferred from a short caption. We also try to isolate object detection capability from spatial reasoning ability by simplifying objects to symbols.
> 4. **Different Interpretation of Visual Blindness**: While the visual blindness of VLMs has been long studied, the context differs in our benchmark. In typical VQA tasks, bias might stem from the questions themselves [2]. For example, the answer to "What is the color of the grass?" is usually "Green", allowing models to answer the question correctly without seeing the image. Multiple prior works have focused on addressing such biases with new benchmarks [3]. In contrast, none of the questions in our benchmark can be answered without looking at the image and the blindness is unrelated to the question.
>
>
> > *W2 & Q2: The synthetic nature of the tasks introduces artifacts. Good spatial reasoning ability on this task may not generalize to real ones. Any discussions about the synthetic-real gap?*
>
> Thanks for your comments! It is valuable to extend the scope of our work and validate our results on real images. We found that a very recent work [7] released a Densely Captioned Images (DCI) dataset where each image has a detailed caption with more than 1000 words on average. However, this dataset does not include questions. We then carefully curated multiple-choice questions regarding spatial-reasoning (object counting, relation, and position understanding) and annotated the answers. We name this new dataset as Spatial-Real.
>
>
> The evaluation results are shown in **Table 1**  in the additional PDF. The same trends still hold for real images (see VQA vs. VTQA, TQA (LLM) vs. VTQA, TQA (LLM) vs. VQA in **Table G.2**). In addition, compared to Fig 4 and 10 in the paper, the overall accuracy increases across all three input modalities (TQA, VQA, VTQA) in the Spatial-Real benchmark. However, the modality gap (accuracy difference between VTQA and VQA) grows from 7.0% on synthetic benchmarks (avg) to 30.0% (avg) on Spatial-Real.
>
> We will open source this new dataset and the full results will be included in our revised manuscript.
>
> > *Q3: In Spatial-Map, will VLMs describe directions using “left/right/up/down”?*
>
> Great point! We carefully examined the VLMs' responses and did not observe notable differences in the way they describe spatial directions. This assessment was done by manually checking all results.
>
> Given that the options with directions (e.g., A. Northwest, B. Southwest, C. Southeast, D. Northeast) are included in the question prompt, we observed consistent adherence to these instructions across all models. Rather than using relative terms like "left/right/up/down," the models consistently responded with the specified cardinal directions.
>
> Examples of typical responses include:
> - "A. Northwest"
> - "Children's Choice Toys is located to the northeast of Yak Yarns."
> - "The correct option is B. Northeast."
>
>
> [6] Zhang et al., MathVerse: Does Your Multi-modal LLM Truly See the Diagrams in Visual Math Problems?
>
> [7] Urbanek et al., A Picture is Worth More Than 77 Text Tokens: Evaluating CLIP-Style Models on Dense Captions, CVPR 2024.

---

> > ### Comment · Reviewer_i8Xy · 2024-08-13
> > **Thanks for the response**
> >
> > I thank the authors for providing the rebuttal. The rebuttal discusses the differences with prior works. I also appreciate the effort for experimenting with real dataset - it is interesting to see that the modality gaps becomes even larger on real image.
> >
> > I am still concerned about the artificially created toy tasks in the paper. I think more experiments with real data, or semi-real data like rendered images using graphics (e.g. CLEVR) that are 3D, will make the paper much stronger.
> >
> > I will keep my score as 5.

---

> > > ### Author Response · Authors · 2024-08-14
> > >
> > > Dear Reviewer i8Xy,
> > >
> > > Thank you for affirming our rebuttal, positive recommendation, and valuable suggestions! We acknowledge the importance of real datasets and are committed to expanding both the diversity and scale of Spatial-Real. While we recognize the value of natural data, we also believe that synthetic tasks serve an important role. As acknowledged by Reviewer JHoY, these tasks can serve as cognitive tests that evaluate basic capabilities, which are relevant to broader real world applications. This approach has precedence in fields like IQ testing, where evaluating foundational cognitive skills is valuable.
> > >
> > > We will open source our benchmark to facilitate future research in this area.
> > >
> > >
> > > Best,
> > >
> > > Authors

---

### Official Review · Reviewer_kVxr · 2024-07-13

**Soundness:** 2
**Presentation:** 3
**Contribution:** 2
**Rating:** 4
**Confidence:** 4

**Summary:**

This paper develops a novel benchmark to understand spatial reasoning ability of LLM and VLM. Using such a benchmark, authors conduct experiments to evaluate models' performance, and reveal several results.

**Strengths:**

1. This paper is well-written and well-structured.

2. This paper conducts a series of experiments and include very recent LLMs and VLMs.

**Weaknesses:**

1. The design rationales of the three benchmarks are not included or discussed. "Spatial-Map" and "Maze-Nav" look reasonable but "Spatial-grid" seems ill-posed. Would we really meet any similar application scenario in the real world?

2. In the experiments of "The impact of input modality", LLMs and VLMs take very different inputs in the benchmarks, with no part overlapping. In this case, it is unsure whether such a comparison is fair or even valid.

3. It is unclear why the results of "Spatial Map" are not provided in main paper or appendix. In addition, the performance of some VLMs are also missing. Given that it can be already observed that there exist some outliers, it is important to show full results of the study to ensure the validity of observations.

4. LLaVA-1.6-34B is the most performant model in this model family. However, in Figure 7-9, it consistently under-performs compared to other LLaVA models on Spatial grid. I would like to see authors' comments on these finding. What might be the cause to the weakness of LLaVA-1.6-34B?

5. This paper overall provides quantitative results of current VLMs without deeper analysis. It remains unclear what on earth causes the weakness of these VLMs. What are insights or takeaway to improve VLMs?

**Questions:**

See weakness.

**Limitations:**

No negative societal impacts of the study.

---

> ### Author Rebuttal · Authors · 2024-08-07
>
> We sincerely appreciate your comments and questions, which we address in detail below.
>
> > *Q1: Design rationales of the three benchmarks. Real world application scenarios for "Spatial-Grid"?*
>
> Our benchmarks are designed to cover diverse aspects of spatial reasoning, such as spatial relationships, navigation, position understanding, and object counting (L34-42). We illustrate the design rationale of each benchmark in more detail:
> - Spatial-Map: This benchmark resembles a map scattered with numerous objects (e.g., hotels and stores), each represented by distinct symbols. It evaluates the model's ability to identify and understand the spatial relationships between these dispersed objects.
> - Maze-Nav: This benchmark simulates navigation scenarios. It evaluates the model's capability to understand and navigate through complex environments, akin to finding a path in a maze.
> - Spatial-Grid: This benchmark reflects scenarios with dense visual information in structured, grid-like environments where understanding the **layout** is crucial.
>
> We highlight several real-world applications for Spatial-Grid:
> - Warehouse Manipulation: Autonomous robots in warehouses navigate through grid-like storage layouts where objects are densely arranged. These robots need precise spatial understanding to efficiently retrieve and restock items, making them highly dependent on reliable spatial reasoning capabilities similar to those tested in Spatial-Grid.
> - Traffic Monitoring: Systems that identify and count vehicles within structured scenes, such as busy intersections with vehicles arranged compactly in lanes and rows, rely on accurate localization and counting. Such capabilities are critical for the safe deployment of VLM-based systems in traffic management.
> - Diagram and Document Understanding: An emerging application for MLLMs is understanding documents that contain dense collections of visual elements arranged in structured layouts. As businesses and educational sectors increasingly rely on digital data, the ability to parse and understand complex documents becomes crucial.
>
> > *Q2: Fairness of the experiment on "The impact of input modality" as "LLMs and VLMs take very different, non-overlapping inputs in the benchmarks"*
>
> It is indeed challenging to directly compare LLMs and VLMs due to their difference in input modality. This is exactly the purpose of our benchmarks: for each sample, we create semantic overlap between the textual description and the image, where the input in each modality has similar information sufficient to answer the question.
> Instead of comparing LLM vs. VLM, we compare TQA (LLM), TQA (VLM), VQA, and VTQA. We would like to clarify the terminologies and input modalities considered for different models, summarized in **Table G.1** in the General Response.
> To investigate "The impact of input modality", we conducted multiple set of controlled experiments between TQA, VQA, and VTQA. As acknowledged by R2 (i8Xy), "The TQA, VQA, VTQA test is a smart way to reveal the modality bias in VLMs and LLMs, clearly showing the blindness of VLMs."
> Our comparisons and references in the paper are summarized in **Table G.2** in the General Response.
>
> > *Q3: Missing results of Spatial-Map and some VLMs in the main paper*
>
> Thanks for the question! The results of the "Spatial-Map" benchmark are provided in the main paper in Sec 4 (Fig 4 and 5), with further details available in the ablation studies in Sec 5.2 and 5.3 for both open-sourced and proprietary models (Fig 10, 11, and 12). Additional detailed results can be found in Appendix D.
> In Sec 5.1, the ablation study comparing VTQA vs. TQA (VLM) does not include results for the InstructBLIP family because these models do not support using only their LLM backbone. We apologize for the oversight of not including the "Spatial-Map" results in Sec 5.1. This was an unintentional mistake. The full results have been included in the Appendix, and we will ensure they are added to the main paper in the revised manuscript.
>
> > *Q4: Why LLaVA-1.6-34B underperforms other LLaVA models on Spatial-Grid?*
>
> Great point! Fig 4, 5, 10, and 11 show that while LLaVA-1.6-34B consistently outperforms other LLaVA models on Spatial-Map and Maze-Nav, it lags in Spatial-Grid.
> We conducted an ablation study, where we added three more questions Q4-Q6 (object in bottom-right, top-right, bottom-left corners) besides Q1-Q3 in Table 5 (Appendix D).
>
> Detailed breakdowns can be found in **Table 2** in the additional one-page PDF. The results indicate that LLaVA-1.6-34B excels in counting (Q1) but struggles with layout and fine-grained visual interpretation (Q2-Q6), limiting its spatial reasoning in dense grid environments. However, we believe a deeper understanding of this phenomenon is beyond the scope of our work. We hope our benchmark serves as a springboard for further study on spatial reasoning and design of better reliable MLLMs.
>
> > *Q5: It remains unclear what on earth causes the weakness of these VLMs. Insights or takeaway to improve VLMs?*
>
> Due to the differences in model architecture, training pipelines, the scale and diversity of training data, pinpointing the precise causes of weaknesses in modern VLMs is fundamentally challenging and beyond the scope of our study.
> The primary purposes of this paper include: (1) Introducing a pioneering benchmark that evaluates diverse aspects of spatial reasoning with multimodal inputs. (2) Conducting a timely and comprehensive evaluation of a wide range of both open-sourced and proprietary LLMs and VLMs.(3) Highlighting the current limitations of VLMs in spatial reasoning, thereby setting the stage for further investigations.
> In Sec 6, we proposed several hypotheses to explain the observed discrepancies among VLMs. While we acknowledge understanding the 'why' behind performance differences is crucial, we believe first systematically documenting and decomposing the observed phenomena is equally valuable to the research community.

---

### Author Rebuttal · Authors · 2024-08-07

**Review summary** We sincerely appreciate all reviewers for their time and effort in providing valuable feedback and suggestions on our work. We are glad that reviewers recognize our work to be _novel_ (R1, R4), _highly impactful_, and _intruguing_ (R4). Additionally, reviewers found our results and analysis _intensive_ (R2) and _interesting_ (R3), with useful insights (R3, R4). We appreciate the positive remarks on our manuscript being well-written and well-structured (R1, R4), and the appreciation for the direction of our research (R3).


We have addressed the comments and questions in individual responses to each reviewer. Below, we include two tables that are frequently referenced in our responses. The attached PDF contains an illustration of our new real-world benchmark **Spatial-Real**, evaluation results (Table 1), and an ablation study for the LLaVA family (Table 2, [R1]).

|Model|Input Modality|Term|Description|
|-|-|-|-|
|LLM| Text-only | TQA (LLM) | Input is purely textual and contains all necessary information of an image to answer the question.|
|VLM|Text-only| TQA (VLM) | Input is purely textual, but applied to VLMs (such as the LLaVA family). In the paper, this setting is called Text-only input with VLM (No Img) (Sec 5.1, Fig 7, Fig 11). We have name it as TQA (VLM) in the revised manuscript for easier reference.|
|VLM|Vision-only|VQA| Image-only input without an equivalent textual description |
|VLM|Vision-text|VTQA| Input includes both an image and its textual description|

#### Table G.1: Terminology and input modalities for LLMs and VLMs.

|Comparison|Results and Analysis|Summary of Findings|
|-|-|-|
|TQA (LLM) vs. VQA|  Sec.4 Figure 5| VLMs (with image-only input) rarely enhance the performance compared to their LLM counterparts (with text-only input).|
|VTQA vs. TQA (VLM)|Sec.5.1 Figure 7|VLMs exhibit improved performance in spatial reasoning tasks when the image input is absent.|
|VQA vs. VTQA|Sec.5.2 Figure 10|Given the same image input, additional textual description enhances VLM's performance.|
|TQA (VLM) vs. TQA (LLM)|Sec.5.2 Figure 11 |Multimodal finetuning enhance LLM's spatial reasoning performance.|
|TQA (LLM) vs. VTQA |Appendix C Figure 15| No definitive winner. |

#### Table G.2: Summary of experiments on the impact of input modalities.

---

### Author Response · Authors · 2024-08-10

Dear Reviewers,

Thank you again for your valuable time and insightful comments!

In response, we have provided detailed answers, additional ablation studies, and a new real-world benchmark with thorough experimental results. As we are halfway through the discussion stage, please let us know if your concerns have been addressed. We are more than happy to address any additional questions before the discussion period officially concludes.

Best,

Authors of Paper 8807

---

### Decision · Program_Chairs · 2024-09-25

**Decision:**

Accept (poster)

**Comment:**

This is an interesting analysis paper. The paper develops benchmarks to evaluate spatial reasoning in large language models (LLMs) and vision-language models (VLMs), revealing that VLMs often underperform compared to LLMs, even when additional visual input is provided. An interesting finding is that VLMs without the visual input perform better than VLMs with the visual input and LLMs with text input alone, highlighting the complexity of spatial reasoning in multimodal tasks.

The reviewers raised concerns about the evaluation methods and comparisons, particularly regarding the limited improvement of VLMs when visual inputs are available. However, the authors provided detailed rebuttals, introducing new baselines and conducting additional experiments that effectively clarified the concerns. The discussion around how redundancy between visual and textual information can be leveraged to improve performance also provided valuable insights.

Overall, the AC believes this is a comprehensive study of spatial reasoning in multimodal models, though the conclusion of treating vision as first-class modality might seem obvious to many vision and language researchers. The AC would highly suggest that the authors add more discussion (and potentially some probing experiments) regarding how to improve the spatial reasoning abilities of VLMs, especially when both modalities are available. Exploring ways to optimize the use of visual input for spatial reasoning would significantly strengthen the paper’s contributions to the field.